# When Does Reasoning Matter? A Controlled Study of Reasoning's Contribution to Model Performance

## Abstract

Large Language Models (LLMs) with reasoning capabilities have achieved state-of-the-art performance on a wide range of tasks. Despite its empirical success, the tasks and model scales at which reasoning becomes effective, as well as its training and inference costs, remain underexplored. In this work, we rely on a synthetic data distillation framework to conduct a large-scale supervised study. We compare Instruction Fine-Tuning (IFT) and reasoning models of varying sizes, on a wide range of math-centric and general-purpose tasks, evaluating both multiple-choice and open-ended formats. Our analysis reveals that reasoning consistently improves model performance, often matching or surpassing significantly larger IFT systems. Notably, while IFT remains Pareto-optimal in training and inference costs, reasoning models become increasingly valuable as model size scales, overcoming IFT performance limits on reasoning-intensive and open-ended tasks.

## 1 Introduction

Large Language Models (LLMs) that generate explicit Chains of Thought (CoT) have rapidly become a defining paradigm. The research community is releasing increasingly capable reasoning models, which consistently outperform standard Instruction Fine-Tuned (IFT) counterparts at test time, especially on math, coding, and other reasoning-heavy tasks DeepSeek-AI (2025); OpenAI (2024); Mistral-AI (2025).

Despite rapid progress, we still lack clarity on when explicit reasoning is most beneficial. Both prior evidence and our findings (Figure 1) point to a highly task-dependent picture: reasoning yields substantial gains on math and coding benchmarks where multi-step problem solving is essential (Zhu et al., 2024), but provides only limited improvements on simpler factual or classification tasks (Liu et al., 2024). As Figure 1 shows, these gains concentrate on reasoning-intensive (e.g., `gsm8k`, `aime`) and open-ended tasks, while benefits on general multiple-choice tasks are much smaller or inconsistent.

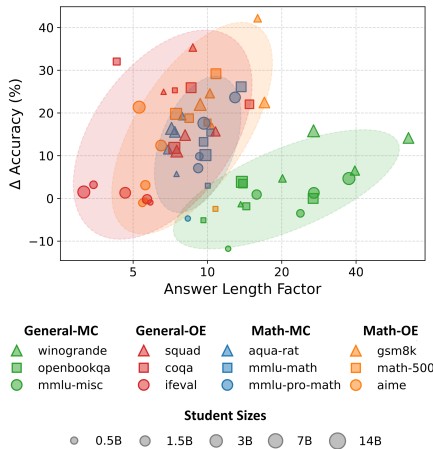

Figure 1: Task sensitivity to reasoning. Reasoning helps most on open-ended and math tasks; gains are limited or inconsistent on general multiple-choice tasks. X-axis: extra-token factor when switching from IFT to reasoning. Y-axis: accuracy gain (%).

Meanwhile, the scaling dynamics of reasoning models pose further challenges. Small models often struggle to absorb the reasoning depth of large teachers unless traces are carefully adapted (Li et al.). Conversely, at larger scales, reasoning appears to unlock performance plateaus that IFT models cannot surpass, as shown by frontier efforts such as OpenAI's o1 reasoning series (OpenAI, 2024) and open-source counterparts like Qwen (Qwen-Team, 2025) and Mistral's Magistral line (Mistral-AI, 2025). While these works emphasize headline results, they don't systematically disentangle

confounding factors such as model scale or training and inference budget, leaving practitioners with little concrete guidance.

The goal of this paper is to bridge these gaps by providing a unified, controlled view of reasoning versus IFT. More broadly, we aim to clarify the design choices shaping reasoning models:

> *Which tasks consistently benefit from reasoning, how do these gains vary with model scale, and how are they balanced against training and inference costs relative to standard IFT?*

**Challenges.** Addressing this question is highly challenging, requiring a controlled experimental setup specifically designed to isolate performance drivers such as data domain, model capacity, and inference budget.

**Our approach.** We investigate this matter with a large-scale, fully controlled distillation setup that holds data and capacity constant while varying the supervision format (IFT vs. reasoning). A single teacher produces paired answers (IFT and reasoning) to the same prompts,[1] enabling like-for-like comparisons across model sizes and domains.

**Contributions.** This paper makes three main contributions:

- **A controlled reasoning testbed for disentangling confounders.** We present a large-scale distillation framework that isolates the effect of supervision format (IFT vs. reasoning) across different model sizes and data domains. This design removes major confounders and enables clean attribution of performance. Using 1.6M IFT-reasoning pairs for training and evaluating over 12 benchmarks (amounting to 70k H100 GPU-hours), we map reasoning's impact across model scale, task family (math vs. general), and answer format (multiple-choice vs. open-ended).

- **Actionable guidance for practitioners.** Reasoning reliably breaks IFT performance plateaus, often matching models several times larger (§ 3), whereas IFT remains a reliably cost-efficient path for both training and inference (§ 4). In a nutshell, reasoning is beneficial when task and scale justify the extra compute, whereas a larger IFT model is preferable otherwise.

- **Open resources.** We release all code and paired training datasets (IFT and reasoning outputs for the same inputs) to enable reproducibility and future controlled studies on reasoning.

## 2 EXPERIMENTAL SETUP

Frontier research initiatives highlight reasoning models' performance but often do not disentangle the underlying sources of improvement, due to opaque data mixtures and shifting supervision schemes. We move the needle by isolating reasoning itself. Using a single teacher that generates paired IFT and reasoning answers to the same prompts, we assess performance across model scales and data domains. This controlled setup enables clean attribution of performance to reasoning while sidestepping the cost of RL pipelines (Mistral-AI, 2025; Qwen-Team, 2025).

### 2.1 FORMALIZATION

**Preliminaries.** We adopt the standard prompt-based generation setting, where a causal language model $f_{\boldsymbol{\theta}} : \Omega^* \to \mathbb{R}^{|\Omega|}$ maps an input text sequence to unnormalized logit scores for next-token prediction. Here, $\Omega = \{\omega_1, \ldots, \omega_{|\Omega|}\}$ is the vocabulary and $\Omega^*$ its Kleene closure.[2] We define the generation mechanism $\mathcal{G}_{\tau,p}$ such that $\mathcal{G}_{\tau,p}(f_\theta) : \Omega^* \to \Omega^*$ represents the recursive generation process of $f_\theta$ under temperature $\tau \geq 0$ and nucleus-sampling parameter $p \in ]0, 1]$. For convenience, we denote this process by $g_\theta$. Intuitively, given a question $\mathbf{x}$, $g_\theta(\mathbf{x})$ corresponds to the answer generated by model $f_\theta$.

---

[1] Examples of data formats are provided in Appendix D.
[2] $\Omega^*$ is the set of all sequences written with elements in $\Omega$. Formally, $\Omega^* = \bigcup_{i=0}^{\infty} \Omega^i$.

**Distillation procedure.** We consider a student model $f_{\boldsymbol{\theta_S}} : \Omega^* \to \mathbb{R}^{|\Omega|}$ and a teacher model $f_{\boldsymbol{\theta_T}} : \Omega^* \times \{0, 1\} \to \mathbb{R}^{|\Omega|}$. Let $g_{\theta_S} : \Omega^* \to \Omega^*$ and $g_{\theta_T} : \Omega^* \times \{0, 1\} \to \Omega^*$ be the generation function of the student and teacher models, respectively. The teacher differs from the student in that it accepts an additional binary input $r \in \{0, 1\}$ indicating whether reasoning mode is enabled ($r = 1$) or disabled ($r = 0$). Given a collection of input questions $X = \{\mathbf{x_i}\}_{i=1}^N$, we construct a synthetic dataset $D = \{(\mathbf{x}_i, g_{\theta_T}(\mathbf{x}_i, r_i))\}_{i=1}^N$, where $r_i \in \{0, 1\}$ specifies whether reasoning is enabled for sample $i$. The distilled student model can be written as $\mathcal{T}_H(f_{\theta_S}, D)$, where $\mathcal{T}_H$ denotes the causal training procedure that updates student $f_{\theta_S}$ on the teacher-generated dataset $D$ under hyperparameters $H$.

## 2.2 Distillation Protocol

**Teacher models** ($f_{\theta_T}$). For data generation, we employ a state-of-the-art open-weight mixture-of-experts model, `Qwen3-235B-A22B` (Qwen-Team, 2025), which includes a configurable flag that enables or disables reasoning mode.

**Student models** ($f_{\theta_S}$). We distill knowledge into five Qwen2.5 base models ranging from 0.5B to 14B parameters: `Qwen-2.5-0.5B`, `*-1.5B`, `*-3B`, `*-7B` and `*-14B` (Yang et al., 2024a; Qwen-Team, 2024). These untuned base checkpoints are chosen from a family distinct from the teachers, reducing pretraining overlap and inductive biases.

**Input questions** ($X$). We consider two regimes that reflect common deployment scenarios. (1) *General-purpose training*: starting from a base student, we distill general teacher capabilities using input questions from the `7M_core` subset of the `Infinity-Instruct` dataset (Li et al., 2025). These questions cover multiple domains, including general knowledge, commonsense Q&A, coding, and math, and are denoted by $X_{\text{general}}$. (2) *Math-centric training*: starting from either a base or a general-distribution-trained student, we distill knowledge on a specific domain. We decide to focus on mathematics, as it is a common reasoning domain. Input questions, $X_{\text{math}}$, are drawn from the `Llama-Nemotron-Post-Training-Dataset` (Bercovich et al., 2025).

**Data generation** ($D$). For each set of input questions $X \in \{X_{\text{general}}, X_{\text{math}}\}$, we generate answers under both $r = 0$ (IFT) and $r = 1$ (reasoning). Formally, $D_{IFT} = \{(\mathbf{x}, g_{\theta_T}(\mathbf{x}, 0)) \mid \mathbf{x} \in X\}$ and $D_R = \{(\mathbf{x}, g_{\theta_T}(\mathbf{x}, 1)) \mid \mathbf{x} \in X\}$. For reasoning generations, we sample with temperature $\tau = 0.6$ and nucleus parameter $p = 0.95$, while for IFT we use $\tau = 0.7$ and $p = 0.8$.[3] In total, to ensure sufficient convergence during model training, we generate 1.6M answer pairs: 1.3M for the general-domain setting and 300K for the math-centric scenario.

**Training** ($\mathcal{T}$). All student models are trained exclusively on synthetic data produced by the teacher; no reinforcement learning is involved. To control the impact of supervision format, we vary the fraction of reasoning versus IFT instances. Let $X_\rho \subseteq X$ be a subset of prompts such that $|X_\rho| \approx \rho|X|$, with $\rho \in [0, 1]$ denoting the reasoning ratio. We then construct $D_R^\rho = \{(\mathbf{x}, \mathbf{y}) \mid (\mathbf{x}, \mathbf{y}) \in D_R, \mathbf{x} \in X_\rho\}$ and $D_{IFT}^\rho = \{(\mathbf{x}, \mathbf{y}) \mid (\mathbf{x}, \mathbf{y}) \in D_{IFT}, \mathbf{x} \in X \setminus X_\rho\}$, and train on their union $D_\rho = D_{IFT}^\rho \cup D_R^\rho$. We evaluate $\rho \in \{0, 0.25, 0.5, 0.75, 1\}$ under two settings: (1) *sequential training* ($\mathcal{T}_{\text{seq}}$), where models are first trained on IFT and then reasoning data, and (2) *mixed training* ($\mathcal{T}_{\text{mix}}$), where both are combined from the start. We also study domain-specific adaptation, where general-domain students are further aligned on math-centric data.[4]

## 2.3 Evaluation Methodology

**Benchmarks.** For comprehensive assessment, we evaluate models on a suite of 12 benchmarks covering both general-purpose and mathematical reasoning, across Multiple-Choice (MC) and Open-Ended (OE) formats. For general-purpose MC tasks, we use `winogrande` (Keisuke et al., 2020), `openbookqa` (Mihaylov et al., 2018), and `mmlu-misc`. For general-purpose OE tasks, we use `squad` (Rajpurkar et al., 2016), `coqa` (Reddy et al., 2019), and `ifeval` (Zhou et al., 2023). In the mathematical domain, MC benchmarks include `aqua-rat` (Ling et al., 2017),

---

[3]Generation parameters were sampled according to the `Qwen3-235B-A22B` model recommendations.

[4]Training hyperparameters $H$ are further discussed in Appendix B.

`mmlu-math` (Hendrycks et al., 2021), and `mmlu-pro-math` (Wang et al., 2024b), while OE benchmarks include `gsm8k` (Cobbe et al., 2021b), `math-500` (Lightman et al., 2023), and `aime` (of Problem Solving, 2025). Additional details on task prompting are provided in Appendix D.

**Inference parameters.** We apply standard decoding with temperature $\tau = 1.0$ and nucleus-sampling parameter $p = 1.0$. To mitigate the limited instruction-following capability of base student models, we evaluate them in a three-shot setting, whereas distilled models are evaluated in a zero-shot setting to directly measure distilled behaviors.

**LLM as a judge.** To ensure consistent and reliable evaluation across tasks, we employ `Llama-3.3-70B-Instruct` (Llama3, 2024), `Nemotron-Ultra-253B-v1` (Bercovich et al., 2025) and `GPT-OSS-120B` (OpenAI, 2025) as judge models, with sampling parameters set to $\tau = 0.7$ and $p = 0.95$. We utilize a majority voting framework to assess response correctness (Zheng et al., 2023; Gu et al., 2024; Verga et al., 2024; Saha et al., 2025). The role of the judges is not to score the answers but to verify semantic equivalence against the ground truths, handling poorly formatted outputs that confound exact-match metrics. Further details are provided in Appendix D.3.

## 3 MODEL PERFORMANCE ANALYSIS

We analyze how downstream performance shifts under different training design choices. Specifically, we vary the supervision format (IFT vs. reasoning) across different model scales and data domains (general vs. math). This setup allows us to disentangle the contribution of reasoning traces from confounding factors, to map where reasoning provides reliable gains, and show how these dynamics interact with model size and task type.

### 3.1 MAIN RESULTS

Figure 2 presents overall results on the impact of model scale, training data format, and distribution on downstream performance in a simple mono-phasic setup, where student models are trained on a single data distribution using a single data format.

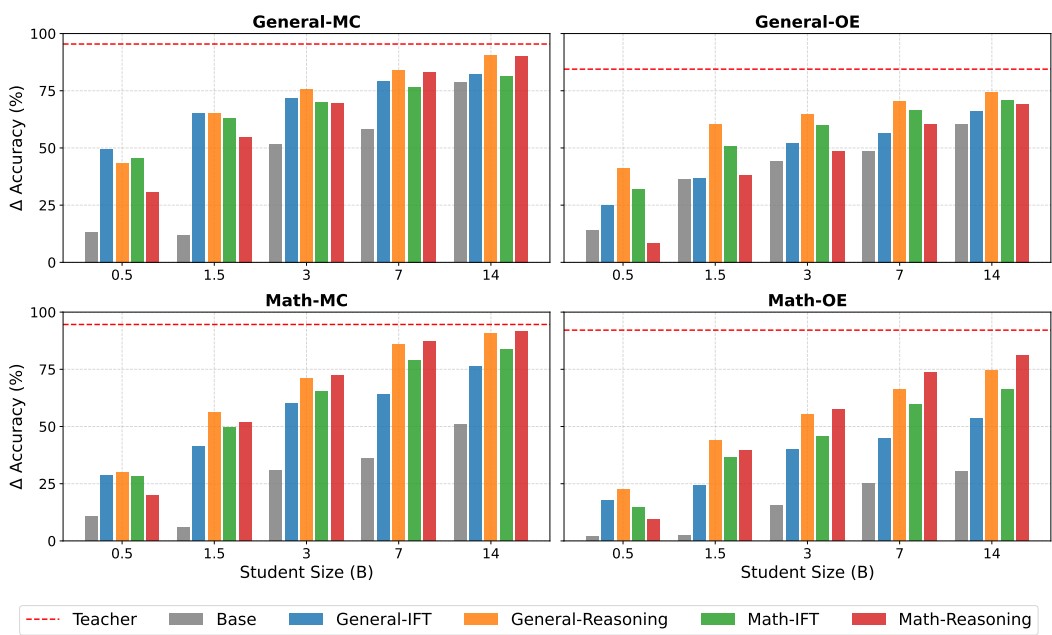

Figure 2: Downstream performance of mono-phasic models. Results are shown for the teacher model and base students, as well as for models trained with IFT- and reasoning-style data on both general and math-centric domains. Additional results and models are provided in Appendix F.

**Reasoning data boosts downstream performance in general distribution training, especially as model scale increases.** Student models trained on a general data distribution with reasoning globally achieve higher accuracies across benchmarks compared to those trained with IFT. Specifically, on General-OE, Math-OE, and Math-MC tasks, reasoning enables 3B students to match or closely approach the accuracy of 14B IFT models, demonstrating robust accuracy gains from reasoning. An exception occurs on General-MC tasks, where reasoning provides less consistent benefits, and IFT data remains competitive for models under 1.5B parameters, suggesting that smaller models struggle to exploit reasoning data on less reasoning-intensive tasks.

**Math-centric training helps large models on the most reasoning-intensive tasks.** Similar to general-distribution training, the benefits of reasoning on math-centric data increase with model scale, though they exhibit distinct patterns across task categories. For non-math downstream tasks, reasoning data surpasses IFT on General-MC solely for models of 7B parameters or more; on General-OE, it establishes parity at the 14B scale. In contrast, on mathematical tasks, the advantage of reasoning data over IFT emerges at lower scales (around 1.5B). Notably, math-specialized reasoning models achieve comparable performance to general-distribution training once model size exceeds 3B for math tasks, 7B for General-MC, and 14B for General-OE, despite using only a quarter of the training samples (300K versus 1.3M). Overall, this suggests that while larger models gain the most from math reasoning traces, smaller models should continue to additionally rely on general-distribution training to maximize performance across tasks, even over domain-specific distributions.

## 3.2 Impact of Mixing IFT and Reasoning Data

Motivated by the strong performance of reasoning models, we further investigate their effectiveness by varying the proportion of reasoning instances in the general training mix. Specifically, we examine potential synergies between IFT and reasoning under both the sequential and mixed approaches ($\mathcal{T}_{\text{seq}}$ and $\mathcal{T}_{\text{mix}}$, respectively; see § 2), and subsequently analyze scaling behaviors in sequential training relative to the reasoning ratio and model size.

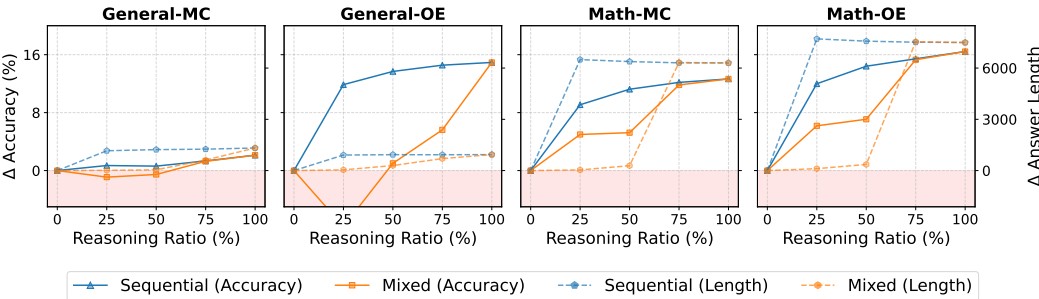

Figure 3: Comparison of sequential and mixed training scenarios across varying reasoning ratios. The accuracy gap relative to the IFT baseline (0% ratio) is shown with solid lines, while the average answer length (in tokens) is reported with dashes. Results are averaged over all student sizes.

**Mixed training exhibits moderate IFT-reasoning synergies.** We motivate our analysis of mixed training by the hypothesis that models can acquire reasoning abilities while retaining the conciseness of IFT-style answers. Figure 3 confirms that, for math tasks, mixed training with a 25–50% reasoning ratio significantly outperforms pure IFT while keeping responses concise, indicating some IFT-reasoning synergy. However, mixed training exhibits pronounced instability, as evidenced by higher variance in accuracy across reasoning ratios (most notably on General-OE). Additionally, models tend to transition abruptly into reasoning mode once reasoning instances exceed 50% of the training mix, suggesting that they adopt reasoning-style outputs whenever the majority of training data is reasoning-focused. In consequence, we focus on the sequential setting for the remainder of this study, leaving stabilization of mixed-style training and consistent exploitation of its potential benefits to future work.

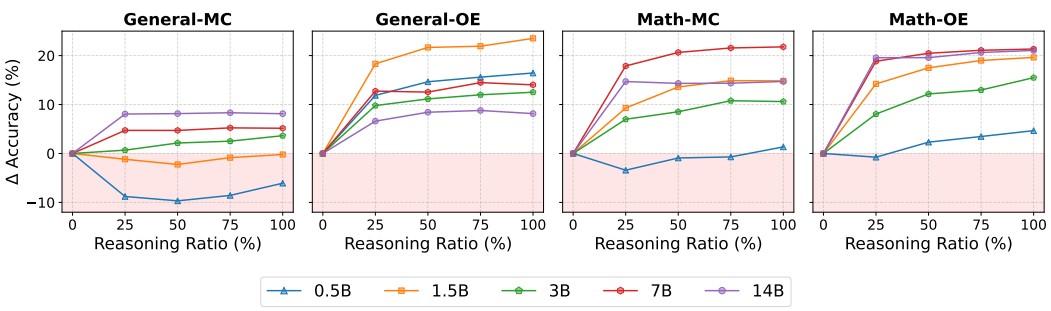

Figure 4: Impact of the reasoning ratio on downstream performance. Results show the accuracy gap relative to the IFT baseline (0% reasoning ratio) in the sequential training scenario, where models are first trained on IFT- and then on reasoning-style data.

**Sequentially combining IFT and reasoning yields no accuracy gains.** Consistent with prior work (Mistral-AI, 2025), Figure 4 shows that "cold-start" training with IFT data (ratios of 25%, 50%, and 75%) does not boost performance. The sole exception is the 0.5B model on General-MC tasks, where IFT-only achieves the highest accuracy.

**Open-ended tasks benefit the most of reasoning.** Varying the reasoning ratio reveals two distinct patterns depending on the downstream task family (Figure 4). For multiple-choice tasks, accuracy plateaus as the reasoning ratio increases (25% for General-MC and 75% for Math-MC), indicating limited benefit from further reasoning-based training. In contrast, for open-ended tasks, especially Math-OE, accuracy continues to rise with higher reasoning ratios across all student sizes, suggesting headroom for extended reasoning training.

## 3.3 DOMAIN-SPECIFIC ADAPTATION

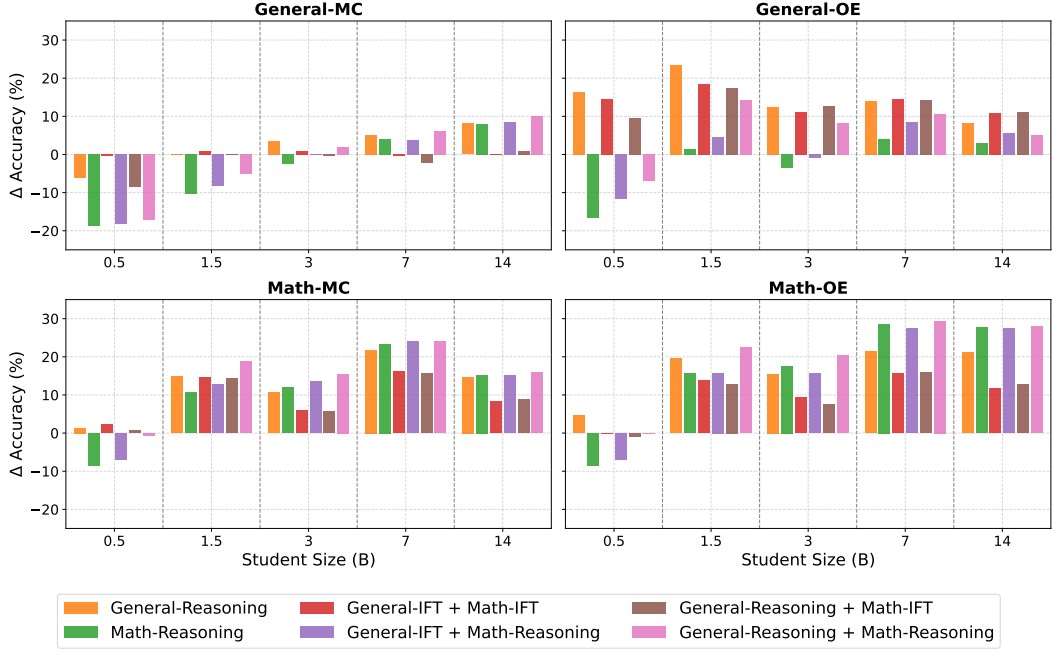

Figure 5: Downstream performance of models trained sequentially on general and math-centric data. Results show the accuracy gap relative to mono-phasic general-domain IFT models (General-IFT in Figure 2). Mono-phasic reasoning models are included as baselines.

In this subsection, building on established training practices, we study bi-phasic strategies in which models are further trained on a targeted domain starting from checkpoints pretrained on general-distribution data (Bolton et al., 2024; Alves et al., 2024; Shao et al., 2024a; Yang et al., 2024b).

**IFT adaptation of a reasoning model provides no benefit.** Applying IFT alignment on a model that has already performed general-reasoning training results in performance that is at best comparable to two-stage IFT, and often worse for smaller models (Figure 5). We observe no positive interaction between reasoning and subsequent IFT adaptation; in some cases, performance even declines relative to general-reasoning models, consistent with the findings reported in §3.2.

**Domain-specific alignment yields performance gains at larger model scales.** Math-centric adaptation can yield significant performance gains, but only under specific conditions. Models with 1.5B parameters and above, particularly when initialized from a general-distribution reasoning checkpoint fine-tuned on a math-centric distribution, achieve the strongest results on mathematical tasks. Under the same setup, models beyond 3B parameters not only match the performance of exclusively math-specialized models but also maintain their non-specific reasoning capabilities, demonstrating an ideal balance between improved in-domain results and robust general-purpose abilities. In contrast, models below 1.5B parameters exhibit signs of catastrophic forgetting (Kirkpatrick et al., 2017) under the same adaptation regime, with 0.5B student even experiencing a global drop in performance, indicating insufficient capacity to solve challenging reasoning tasks.

## 4 ACCURACY-EFFICIENCY TRADE-OFF ANALYSIS

Reasoning outputs are typically longer than IFT responses, making both training and inference more expensive. In this section, we move beyond raw accuracy to analyze the accuracy–efficiency trade-off. All results are reported for general-distribution training from base checkpoints.

### 4.1 TRAINING EFFICIENCY

We first contextualize accuracy relative to training compute (Figure 6). In a sequential distillation setup, we vary the proportion of reasoning instances to examine the trade-offs between performance and training cost in FLOPs. Accounting details are provided in Appendix C.

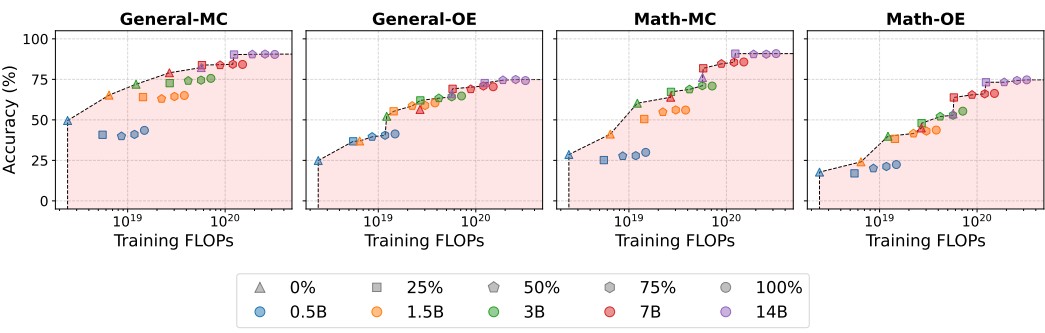

Figure 6: Accuracy versus training FLOPs for models trained with IFT (0%), reasoning-style data (100%), and sequential reasoning ratios of 25%, 50%, and 75%. The Pareto frontier (black dashed lines) highlights efficient configurations, while those that lie in the red-shaded area are suboptimal.

**IFT is an efficient training strategy.** Across all tasks, IFT models follow the Pareto frontier, indicating that scaling model size rather than incorporating reasoning-based training is a reliable approach to achieve performance gains without substantially increasing training costs.

**Reasoning models reach training efficiency as scale increases.** IFT models exhibit an earlier performance plateau compared to models trained with reasoning data, suggesting that additional

gains could be obtained by integrating reasoning into the training mix. In fact, reasoning models ($\geq 25\%$ reasoning ratio) achieve Pareto optimality at larger scales, with some variation across downstream tasks (e.g., 0.5B for General-OE and 7B for General-MC).

**Intermediate reasoning ratios achieve Pareto-optimal trade-offs.** Models trained with a 100% reasoning ratio never reach the Pareto frontier. While sufficiently large models may benefit from improved performance, this comes at the cost of significantly heavier training. In contrast, intermediate ratios (25%, 50%, or 75%) consistently lie on the Pareto frontier, offering controlled performance gains without incurring excessive training cost. This pattern suggests that practitioners should either scale model size or prefer moderate reasoning ratios to optimize the accuracy-efficiency trade-off.

## 4.2 Inference Efficiency

In this subsection, we adopt the perspective of a user leveraging the models for generation purposes. Training is treated as an offline cost, and we evaluate accuracy with respect to inference FLOPs (Figure 7).

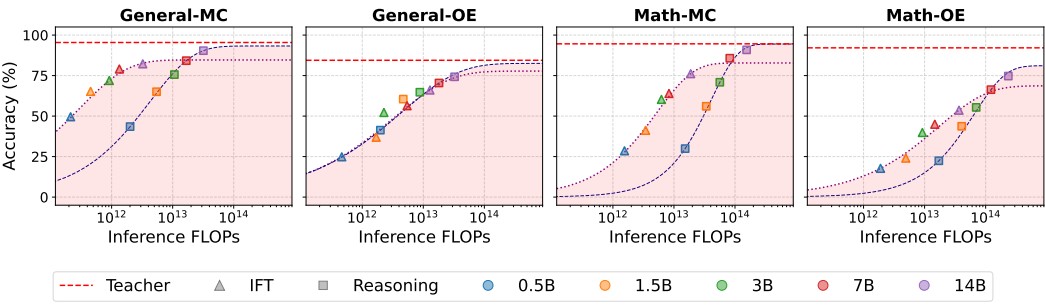

Figure 7: Accuracy versus inference FLOPs for models trained with IFT (0% reasoning ratio) and reasoning-style (100% reasoning ratio) data. The purple-dotted and blue-dashed lines indicate the accuracy-FLOPs interpolated curves for IFT and reasoning, respectively (further details in Appendix E). The red-shaded region highlights configurations that are Pareto-suboptimal.

**IFT is always Pareto-optimal.** Consistent with the observations in § 4.1, IFT models lie on the Pareto frontier across tasks, indicating that increasing model size reliably yields Pareto-optimal gains in inference efficiency.

**Reasoning becomes Pareto-optimal at larger scales.** Trends in the Pareto plots reveal that all reasoning models approach the Pareto frontier as model size increases, with patterns varying depending on the task, while IFT models tend to plateau earlier. This trend is particularly notable for models above 7B, suggesting the benefits of reasoning-based scaling beyond this size. Confirming this hypothesis would require experiments with models larger than 14B parameters, which we leave for future work for practical reasons.

**Open-ended tasks benefit more from reasoning than multiple-choice.** Building on the findings in § 3.2, which show that open-ended tasks gain the most in accuracy from reasoning, we further observe that they also incur smaller relative increases in inference cost compared to multiple-choice tasks. Specifically, switching from IFT to reasoning on open-ended tasks results in an approximate $7\times$ increase in inference cost, whereas for General-MC tasks the increase is around $10$–$15\times$ (see further details in Appendix F, Figure 17). These results support the idea that certain tasks are inherently more reasoning-sensitive, as characterized in Figure 1.

**Longer generations tend to be incorrect.** To gain further insights into inference efficiency, we analyze evaluation-time reasoning traces and find a strong positive correlation between answer length and error rate (Figure 8). In Appendix F (Figure 15), we test a budgeted decoding abstention mechanism that halts generation once a fixed token budget is reached. While this policy reduces inference FLOPs, it substantially decreases accuracy, shifting performance off the Pareto frontier.

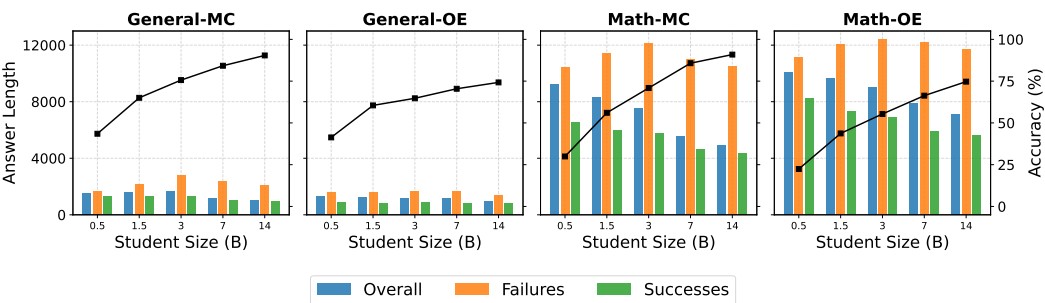

Figure 8: Answer length analysis across student sizes and correctness in reasoning models. Vertical bars indicate average answer lengths for each task category, while the black line shows the corresponding downstream accuracies.

## 5 RELATED WORK

**Instruction tuning and reasoning.** Instruction Fine-Tuning (IFT) has been the standard recipe for aligning LLMs with human instructions (Wei et al., 2022b; Ouyang et al., 2022; Chung et al., 2022). Chain-of-Thought (CoT) extended this paradigm by supervising intermediate reasoning steps, yielding strong gains on arithmetic, symbolic, and commonsense reasoning benchmarks (Rajani et al., 2019; Nye et al., 2021; Cobbe et al., 2021a; Wei et al., 2022a; Kojima et al., 2022). These findings sparked a new wave of reasoning-centric models from both frontier labs and the open-source community. However, most reports highlight aggregate improvements without disentangling when and why reasoning helps, a gap our work addresses.

**Reinforcement learning for reasoning.** Recent frontier efforts extend beyond supervised traces, using Reinforcement Learning (RL) to refine reasoning strategies. Methods such as TRPO (Schulman et al., 2015), PPO (Schulman et al., 2017), and GRPO (Shao et al., 2024b) optimize reasoning trajectories with outcome-based rewards, such as correctness of derivations or code executability (OpenAI, 2024; DeepSeek-AI, 2025; Mistral-AI, 2025). While effective, these methods are compute-heavy and opaque about the precise drivers of performance gains. By contrast, our fully supervised distillation setup isolates reasoning signals without RL, enabling clearer attribution.

**Knowledge distillation.** Knowledge Distillation (KD) transfers capabilities from strong teachers to smaller students (Buciluundefined et al., 2006; Hinton et al., 2015b). Beyond representation-based KD, text-based distillation has become central for reasoning: large teacher models generate either IFT- or reasoning-style traces that guide student learning (Kim & Rush, 2016; Zhou & Chiam, 2023; Hsieh et al., 2023; He et al., 2024). This approach reduces the cost of expensive RL while preserving the performance (DeepSeek-AI, 2025; Qwen-Team, 2025; Mistral-AI, 2025). Yet, prior studies largely focus on showcasing empirical gains rather than dissecting the task- and scale-dependent trade-offs. Our contribution is to turn this distillation pipeline into a controlled testbed, stripping away confounders.

## 6 CONCLUSION

Through a large-scale, distillation-based controlled study, we characterize scenarios when reasoning yields the greatest benefits, showing how its effectiveness depends on model scale, task type, and computational cost. While classical IFT models remain a reliably Pareto-optimal baseline, reasoning consistently delivers substantial gains on open-ended and reasoning-intensive tasks above the 7B-parameter scale, enabling models to break past the performance plateaus of IFT. These results suggest that reasoning signals are not just redundant supervision but a complementary resource that grows in value with scale, pointing toward hybrid approaches that harness reasoning capabilities alongside IFT's conciseness.

## ETHICS STATEMENT

**Environmental and compute considerations.** This work provides an in-depth analysis of scenarios where enabling reasoning capabilities in models is beneficial, as well as where it may not be. In an era where practitioners often prioritize accuracy above all else, we contextualize performance relative to both training and inference costs, offering guidance to avoid excessive computational overhead across different use cases.

**Responsible use of LLMs.** In preparing this manuscript, we occasionally used suggestions from LLMs (GPT-5) to guide improvements in clarity, grammar, and overall readability. All scientific content, including experimental design, codebase, data analysis, results, and interpretations, is independently developed by the authors. LLMs are not involved in generating, modifying, or interpreting any experimental results, nor in producing code or analyses. Their use is strictly limited to selectively refining language to ensure clear and effective communication of our research.

## REPRODUCIBILITY STATEMENT

We have taken every effort to ensure the reproducibility of our experiments. All training and evaluation procedures are described in detail, including the base models, datasets, and all relevant training and generation hyperparameters. To further facilitate replication, we release all project artifacts, including trained models, data generation scripts, training scripts, and evaluation code.

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

## A    Discussion

Several avenues remain for extending our understanding of the conditions under which reasoning distillation is most effective. Future work could explore training dynamics such as convergence behavior (Hoffmann et al., 2022) with respect to dataset size, or assessing larger student models, may help explore potential gains from additional scaling. Other promising avenues include replicating our controlled setup in other scenarios such as reinforcement learning (Schulman et al., 2017; 2015; Shao et al., 2024b), teacher-student logits distillation (Hinton et al., 2015a; Boizard et al., 2024), or exploring alternative techniques beyond SFT, such as preference-based optimization (Rafailov et al., 2024; Xu et al., 2024; Gisserot-Boukhlef et al., 2024b). Additionally, applying data filtering Byrd & Srivastava (2022); AlKhuzaey et al. (2024) to separate examples that require extensive reasoning from those solvable with short, IFT-style responses could resolve the mixed training instability discussed in §3.2.

## B    Training Hyperparameters

All training runs are performed for a single epoch with a global batch size of 262,144 tokens across 16 H100 GPUs. The learning rate follows a Warmup-Stable-Decay (WSD) schedule (Shen et al., 2024) (150-step linear warmup, constant plateau, and 300-step linear decay to 10% of the peak value), using the `AdamW_fused` optimizer (Loshchilov & Hutter, 2019). Peak learning rates are selected via grid search over $\{2\times10^{-5}, 1\times10^{-5}, 7\times10^{-6}, 5\times10^{-6}, 3\times10^{-6}, 1\times10^{-6}\}$. We list in Table 1 the peak learning rates used for student distillation across all models and both data formats (reasoning and IFT). Notably, reasoning-based distillation generally benefits from slightly higher learning rates than IFT.

| Model | Reasoning | IFT |
|---|---|---|
| Qwen2.5-0.5B | 2e-5 | 1e-5 |
| Qwen2.5-1.5B | 1e-5 | 7e-6 |
| Qwen2.5-3B | 7e-6 | 5e-6 |
| Qwen2.5-7B | 5e-6 | 3e-6 |
| Qwen2.5-14B | 3e-6 | 1e-6 |

Table 1: Peak learning rates selected for each student model and training data format.

## C    FLOPs Computation

In this section, we present the methodology used to compute both training and inference FLOPs, following the approach proposed by Hoffmann et al. (2022).

### C.1    Notations

We introduce the following notations for FLOPs computations:

- $V$ : vocabulary size
- $d_{\text{model}}$ : hidden dimension of the model
- $d_{\text{ff}}$ : dimension of feed-forward layers
- $h$ : number of attention heads
- $N_l$ : number of transformer layers
- $l$ : sequence length
- $l_p$ : prompt length
- $l_g$ : generation length
- $N_s$ : number of training samples

## C.2 TRAINING FLOPs

The following formulas compute the FLOPs for model training, assuming a batch size of 1. It is reasonable to assume that the FLOPs are largely independent of the batch size.

$$
\text{FLOPs}_{\text{forward}} = \underbrace{2\,l\,V\,d_{\text{model}}}_{\text{embeddings}} + \underbrace{\left(6\,l\,d_{\text{model}}^2 + 2\,l^2\,d_{\text{model}} + 3\,l^2\,h + 2\,l^2\,d_{\text{model}} + 2\,l\,d_{\text{model}}^2\right) \cdot N_l}_{\text{attention}}
$$
$$
+ \underbrace{4\,l\,d_{\text{model}}\,d_{\text{ff}}\,N_l}_{\text{feed-forward}} + \underbrace{2\,l\,d_{\text{model}}\,V}_{\text{output logits}}
$$
(1)

$$
\text{FLOPs}_{\text{training step}} = 3 \cdot \text{FLOPs}_{\text{forward}}
$$
(2)

$$
\text{FLOPs}_{\text{training}} = \sum_{i=1}^{N_s} \text{FLOPs}_{\text{training step}}(i)
$$
(3)

## C.3 INFERENCE FLOPs

The following formulas compute the FLOPs for model inference. $\text{FLOPs}_{\text{inference}}$ and $\text{FLOPs}_{\text{inference with cache}}$ correspond to single-token generation. $\text{FLOPs}_{\text{inference with cache}}$ assumes that past token keys and values are stored in memory and do not need to be recomputed.

$$
\text{FLOPs}_{\text{inference}} = \underbrace{2\,l_p\,d_{\text{model}}\,V}_{\text{embeddings}} + \underbrace{\left(6\,l_p\,d_{\text{model}}^2 + 2\,l_p^2\,d_{\text{model}} + 3\,l_p^2\,h + 2\,l_p^2\,d_{\text{model}} + 2\,l_p\,d_{\text{model}}^2\right) \cdot N_l}_{\text{attention}}
$$
$$
+ \underbrace{4\,l_p\,d_{\text{model}}\,d_{\text{ff}}\,N_l}_{\text{feed-forward}} + \underbrace{2\,d_{\text{model}}\,V}_{\text{output logits}}
$$
(4)

$$
\text{FLOPs}_{\text{inference with cache}} = \underbrace{2\,d_{\text{model}}\,V}_{\text{embedding}}
$$
$$
+ \underbrace{\left(6\,d_{\text{model}}^2 + 2\,l_p\,d_{\text{model}} + 3\,l_p\,h + 2\,l_p\,d_{\text{model}} + 2\,d_{\text{model}}^2\right) \cdot N_l}_{\text{attention}}
$$
$$
+ \underbrace{4\,d_{\text{model}}\,d_{\text{ff}}\,N_l}_{\text{feed-forward}} + \underbrace{2\,d_{\text{model}}\,V}_{\text{logits}}
$$
(5)

$$
\text{FLOPs}_{\text{multi-token inference}} = \text{FLOPs}_{\text{inference}}(l_p) + \sum_{i=1}^{l_g} \text{FLOPs}_{\text{inference with cache}}(l_p + i)
$$
(6)

# D PROMPTING DETAILS

## D.1 DATA GENERATION

### D.1.1 INFINITY-INSTRUCT

| Question | Answer |
|---|---|
| Lotto 6/49 is one of three national lottery games in Canada. Launched on June 12, 1982, Lotto 6/49 was the first nationwide Canadian lottery game to allow players to choose their own numbers. Previous national games, such as the Olympic Lottery, Loto Canada and Superloto used pre-printed numbers on tickets. Lotto 6/49 led to the gradual phase-out of that type of lottery game in Canada. Based on the paragraph above can we conclude that Canada's Lottery 6/49 was very popular in the 70s? | **IFT:** No, we cannot conclude that "Canada's Lottery 6/49 was very popular in the 70s" based on the paragraph. The text states that Lotto 6/49 was launched on June 12, 1982, which is after the 1970s, so the statement is incorrect. <hr> **Reasoning:** `<think>` Okay, let's see. The user is asking if we can conclude that Canada's Lotto 6/49 was very popular in the 70s based on the given paragraph. First, I need to check the dates mentioned. 

 . . . 

 The answer should state that the conclusion is incorrect because the game started in 1982, which is after the 70s.`</think>` 
 No, we cannot conclude that "Canada's Lottery 6/49 was very popular in the 70s" based on the paragraph. The text explicitly states that Lotto 6/49 was **launched on June 12, 1982**, which is in the early 1980s, not the 1970s. The paragraph also notes that earlier national lottery games (e.g., Olympic Lottery, Loto Canada, Superloto) used pre-printed numbers and were phased out after Lotto 6/49's introduction. There is no information provided about the popularity of any lottery games in the 1970s, nor was Lotto 6/49 even in existence during that decade. The conclusion is therefore unsupported. |
| I know that the answer to the question What are isolators found in microwave antennas made of? is in A necessary condition for the aforementioned reciprocity property is that the materials in the antenna and transmission medium are linear and reciprocal. Most materials used in antennas meet these conditions, but some microwave antennas use high-tech components such as isolators and circulators, made of nonreciprocal materials such as ferrite. These can be used to give the antenna a different behavior on receiving than it has on transmitting, which can be useful in applications like radar. Can you tell me what it is? | **IFT:** The isolators found in microwave antennas are made of **nonreciprocal materials such as ferrite**. <hr> **Reasoning:** `<think>` Okay, let's see. The user is asking about what isolators found in microwave antennas are made of. They provided a passage that should contain the answer. First, I need to read through the passage carefully. The key part here is where it mentions isolators and circulators. The sentence says, "made of nonreciprocal materials such as ferrite." So the answer should be ferrite. Wait, but let me make sure. The passage starts by talking about reciprocity in antennas, but then introduces isolators and circulators as exceptions. It says these components are made of nonreciprocal materials. The example given is ferrite. So yes, the answer is definitely ferrite. I should check if there are any other materials mentioned, but the text only lists ferrite as an example. The user might be looking for that specific term. No other materials are named here. So the answer should be ferrite.`</think>` 
 Answer: The isolators found in microwave antennas are made of **ferrite**, a nonreciprocal material. |

Table 2: Examples of generated data from the `Infinity-Instruct` dataset, showing pairs of IFT and reasoning answers for a common query. We report shortened reasoning traces with . . . due to their length.

### D.1.2 LLAMA-NEMOTRON-POST-TRAINING-DATASET

| Question | Answer |
|---|---|
| Solve the following math problem. Make sure to put the answer (and only the answer) inside \boxed{}. Evaluate the limit $$\lim_{n \to \infty} \left( \frac{n-10}{n+1} \right)^{3n+1}$$ | **IFT:** We are asked to evaluate the limit: $\lim_{n \to \infty} \left( \frac{n-10}{n+1} \right)^{3n+1}$. Step 1: Simplify the base expression $\frac{n-10}{n+1} = \frac{n\left(1 - \frac{10}{n}\right)}{n\left(1 + \frac{1}{n}\right)} = \frac{1 - \frac{10}{n}}{1 + \frac{1}{n}}$. $\ldots$ $\left(3 + \frac{1}{n}\right) = -33$. Now exponentiate: $L = e^{-33}$. Final Answer: \boxed{e^{-33}} 

 **Total characters:** 1,417 
<hr> **Reasoning:** \<think\> Okay, so I need to evaluate the limit as $n \to \infty$ of $\left( \frac{n-10}{n+1} \right)^{3n+1}$. Hmm, let's see. I remember that limits involving expressions raised to variables can sometimes be solved using the exponential function and $\ldots$ Thus: $\ln L \approx \lim_{n \to \infty}(3n+1) \cdot \left(-\frac{11}{n+1}\right) = -11 \cdot \lim_{n \to \infty} \frac{3n+1}{n+1} = -11 \cdot 3 = -33$. Exponentiating both sides: $L = e^{-33}$. Final Answer: \boxed{e^{-33}} 

 **Total characters:** 11,225 |
| Solve the following math problem. Make sure to put the answer (and only the answer) inside \boxed{}. Find the number of integer solutions to the equation $xyz = 2003$. | **IFT:** We are asked to find the number of integer solutions to the equation $xyz = 2003$. Step 1: Prime Factorization. First, factor 2003: $2003 \div 17 = 117.823$ (not divisible by 17). $\ldots$ Try dividing by small primes up to $\sqrt{2003} \approx 44.76$. Eventually, we find: \boxed{12}. 

 **Total characters:** 2,806 
<hr> **Reasoning:** \<think\> Okay, so I need to find the number of integer solutions to the equation $xyz = 2003$. Hmm, let me start by understanding what exactly the problem is asking for. We need all ordered triples of integers $(x, y, z)$ such that when you multiply them together, you get 2003. $\ldots$ Thus, only the above 12 combinations are valid. Final Answer: \boxed{12}. 

 **Total characters:** 26,747 |

Table 3: Examples of generated data from the `Llama-Nemotron-Post-Training-Dataset`, showing pairs of IFT and reasoning answers for a common query. We report shortened answers with ... and their respective character counts.

## D.2 EVALUATION PROMPTS

| Benchmark | Instruction |
|---|---|
| gsm8k | Solve the following math problem. Make sure to put the answer (and only answer) inside \boxed{}. |
| math_500 | Solve the following math problem. Make sure to put the answer (and only answer) inside \boxed{}. |
| aime | Solve the following math problem. Make sure to put the answer (and only answer) inside \boxed{}. |
| mmlu_math | Solve the following math problem. Make sure to put the answer (and only answer) inside \boxed{}. |
| mmlu_pro_math | Solve the following math problem. Make sure to put the answer (and only answer) inside \boxed{}. |
| aqua_rat | Solve the following math problem. Make sure to put the answer (and only answer) inside \boxed{}. |
| winogrande | Given a sentence with a blank (_) and two possible options, choose the option that correctly fills the blank so that the sentence makes the most logical sense. Make sure to put the answer (and only answer) inside \boxed{}. |
| openbookqa | Select the option that best completes the scenario based on everyday reasoning about cause and effect. Make sure to put the answer (and only answer) inside \boxed{}. |
| squad | Read the passage and answer the question by selecting the text span from the passage that best answers it. Make sure to put the answer (and only answer) inside \boxed{}. |
| mmlu_misc | Answer the following multiple-choice question by selecting the option that best fits the correct knowledge. Make sure to put the answer (and only answer) inside \boxed{}. |
| coqa | Read the passage and answer the question by selecting the text span from the passage that best answers it. Make sure to put the answer (and only answer) inside \boxed{}. |
| ifeval | Answer the following instruction. |

Table 4: Instruction prompts used for answer generation across evaluation benchmarks.

## D.3 JUDGING PROMPTS AND STATISTICS

| Benchmark | Instruction |
|---|---|
| Default | You will be given a Question, a User Answer (only its ending is shown due to length), and a Ground Truth.
Your task is not to answer the question, but to say if the user answer is equivalent in meaning to the ground truth.

First, extract the final result from both the User Answer and the Ground Truth Answer, based on the Question.
Then, compare the two final results and determine whether they convey the same meaning.
If they are equivalent, respond with \boxed{yes}.
If they are not equivalent, or if the User Answer does not contain a valid answer, respond with \boxed{no}.

Question:
{question}

User Answer:
{answer}

Ground Truth:
{truth} |
| `ifeval` | You will be given an Instruction and a User Answer (only its ending is shown due to length).
Your task is not to answer the Instruction, but to determine whether the User Answer follows all the formal requirements stated in the Instruction. If the User Answer contains a thinking process, you should ignore it and only focus on the final answer.

First, identify every explicit requirement in the Instruction (e.g., no commas, maximum word count, required word occurrences, formatting rules).
Then, compare the User Answer against these requirements.
If all requirements are satisfied, respond with \boxed{yes}.
If any requirement is violated, respond with \boxed{no}.

Question:
{question}

User Answer:
{answer} |

Table 5: Instruction prompts used for LLM-based answer assessment. Default instructions are applied across all benchmarks, except for `ifeval`.

**Median Average Absolute Error.** The median average absolute error between judges across all benchmarks is 1.2. Specifically, the error is 0.9 for the pair `Nemotron-Ultra-253B-v1` and `Llama-3.3-70B-Instruct`, 1.0 for `Nemotron-Ultra-253B-v1` and `GPT-OSS-120B`, and 1.8 for `Llama-3.3-70B-Instruct` and `GPT-OSS-120B`.

## E    DETAILS ON PARETO INTERPOLATION

In §4.2, we show a Pareto plot of accuracy versus inference cost for IFT and reasoning models. To predict the impact of further model scaling on downstream accuracy, we fit a saturating growth interpolation function to the observed data points (Tan & Le, 2019; Kaplan et al., 2020). The objective function is defined as: $f(x) = \alpha + \beta(1 - \exp(-\gamma x^\delta))$, where $x$ denotes the number of FLOPs and $f(x)$ gives the interpolated accuracy. The parameters are subject to the constraints $\alpha, \beta > 0$, $\alpha + \beta$ not exceeding the teacher's accuracy, $\gamma > 0$, and $0 < \delta \le 1$. Intuitively, $f(0) = \alpha$ corresponds to the minimum achievable performance on the benchmark (a random model with 0 FLOPs), while $\lim_{x \to \infty} f(x) = \alpha + \beta$ represents the maximum performance. The parameters $\gamma$ and $\delta$ control the curvature of the interpolated curve. The function is fitted by minimizing the mean absolute error.

## F    ADDITIONAL RESULTS

### F.1    EXTENDED TASK EVALUATION

To broaden our evaluation beyond the general and math domains, we add three additional reasoning-intensive fields: Law, Code, and Physics.

- **Law**: `mmlu-pro-law` (Wang et al., 2024b) (multiple-choice) and `consumer-qa` (Guha et al., 2023) (open-ended).

- **Code**: `mmlu-pro-code` (multiple-choice) and `C-Code-Summarization-Benchmark` (Liu et al., 2021) (open-ended).

- **Physics**: `mmlu-pro-physics` (multiple-choice) and `scibench` (Wang et al., 2024a) (open-ended).

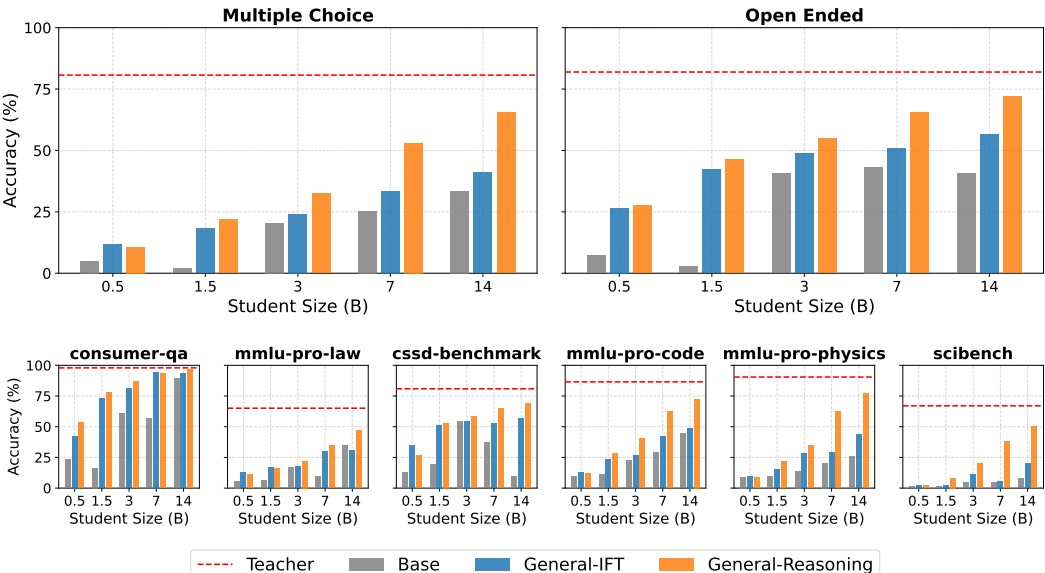

Figure 9: Downstream performance of mono-phasic models evaluated on law, code, and physics tasks. Results are shown for base student models, the teacher, and models trained with IFT- and reasoning-style formats on general-purpose domain.

Consistent with our main analysis (§ 3 - Figure 2), Figure 9 confirms that reasoning supervision outperforms IFT on all benchmarks, with gains scaling positively with model size.

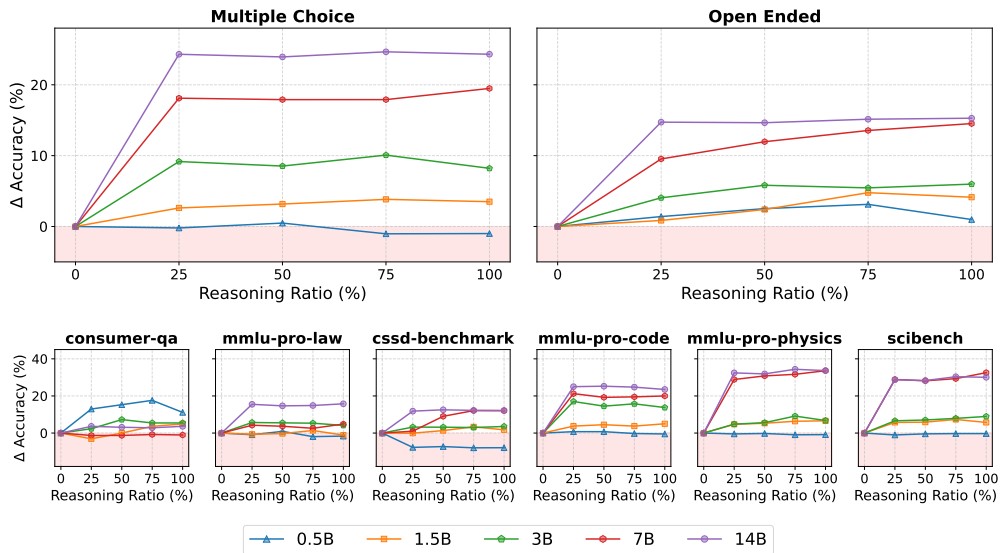

Figure 10: Impact of the reasoning ratio on downstream performance in law, code and physics tasks. Results show the accuracy gap relative to the IFT baseline in the sequential training scenario, where models are first trained on IFT- and then on reasoning-style data.

Extending the primary analysis in §3 and Figure 4, Figure 10 indicates that sequentially combining IFT and reasoning data does not improve performance compared to full-reasoning training. Additionally, as noted in §3.2, open-ended tasks benefit more from larger amounts of reasoning data than multiple-choice tasks.

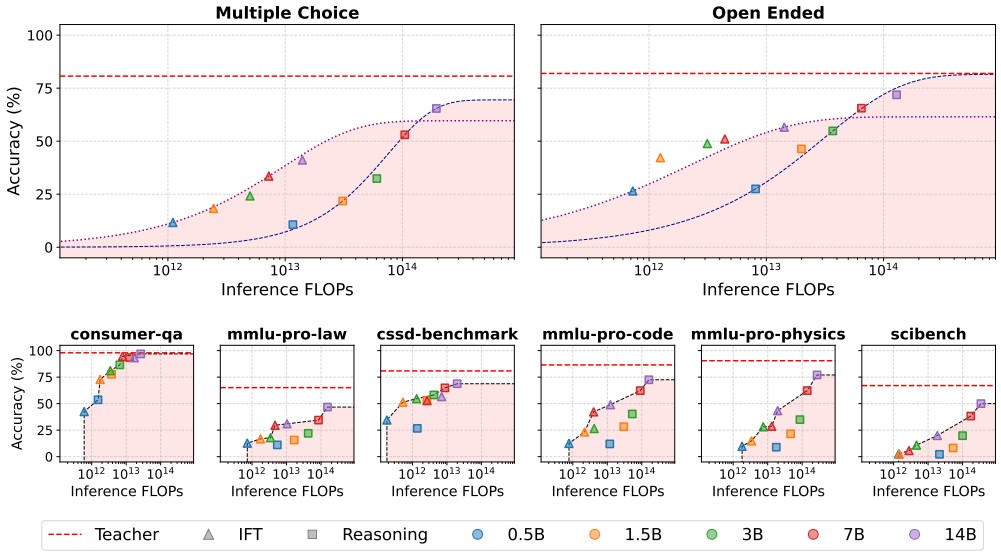

Figure 11: Accuracy versus inference FLOPs for models trained on IFT data (0% reasoning) and reasoning-style data (100% reasoning) across law, code, and physics tasks.

Aligned with §4.2 and Figure 7, Figure 11 shows that reasoning models move closer to the Pareto frontier as model size increases, while IFT models plateau sooner. This effect is particularly noticeable for models above 7B, highlighting the scaling advantages of reasoning-based training at larger sizes.

### F.2 EXPLORING ALTERNATIVE TEACHER–STUDENT CONFIGURATIONS

We extend our study by exploring alternative teacher-student configurations. Specifically, we employ `Nemotron-Super-49B-v1.5` as the teacher paired with `Gemma-3` students (1B, 4B, & 12B). To manage computational costs, we generate 200k paired IFT-reasoning samples from the general-domain set. Downstream evaluation covers general-domain and math-centric benchmarks.

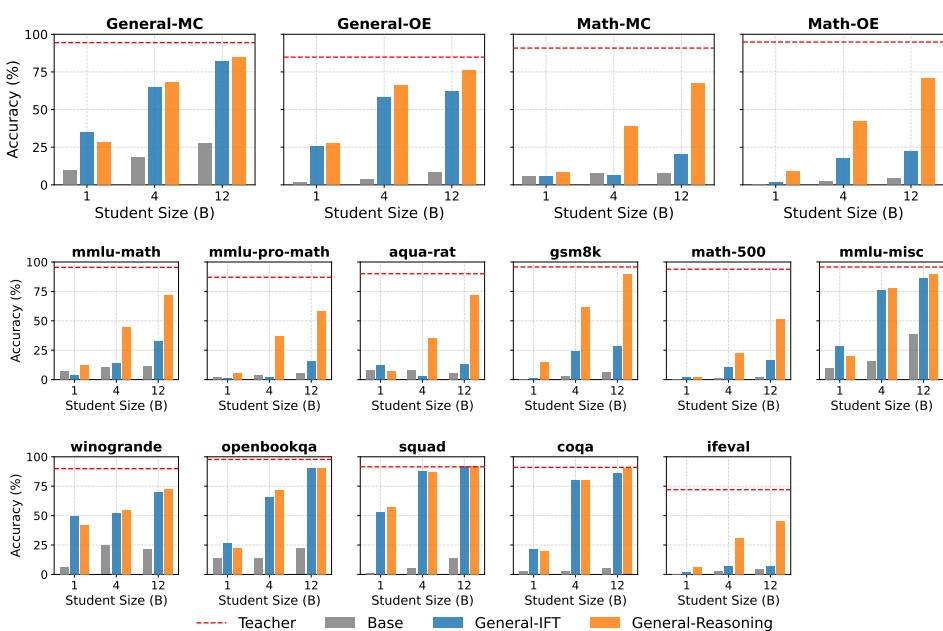

Figure 12: Downstream performance of the `Nemotron-Super-49B-v1.5` teacher with `Gemma-3` students, comparing training on IFT versus reasoning-style data.

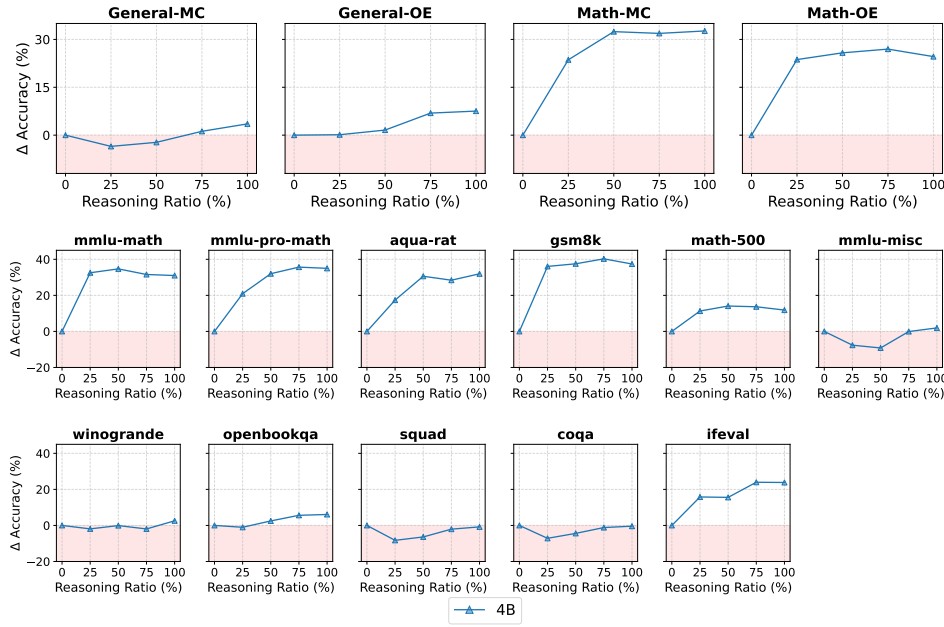

Figure 13: Impact of the reasoning ratio on downstream performance. Accuracy is reported relative to the IFT baseline in the sequential training setup. Results correspond to the Nemotron–Gemma teacher–student configuration at the 4B scale.

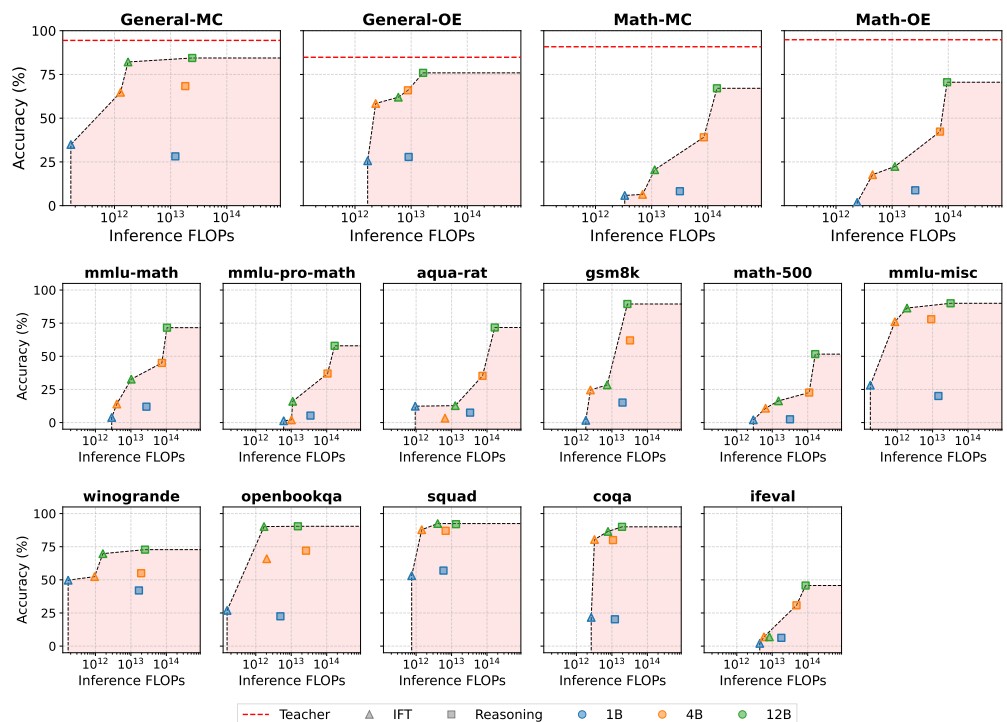

Figure 14: Task-level accuracy versus inference FLOPs for models trained with IFT and reasoning-style data. Results are reported for the Nemotron-Gemma teacher-student configuration.

As observed in § F.1 with varying downstream task domains, experiments with alternative teacher–student configurations do not change the main takeaways. In particular, reasoning-style training continues to outperform IFT as student size increases (Figure 12); sequentially combining IFT and reasoning-style training still does not improve raw downstream performance (Figure 13); and reasoning with small models (below 3B) remains consistently Pareto-suboptimal (Figure 14).

## F.3 GENERATION EARLY-STOPPING

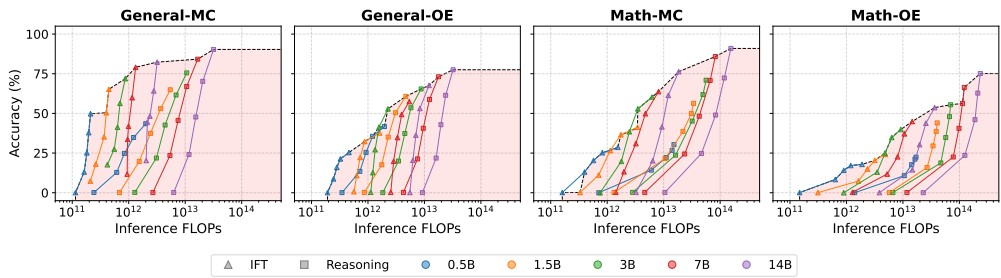

Figure 15: Inference-cost impact of generation early stopping for IFT and reasoning models. Each model is evaluated at five maximum-length thresholds, corresponding to the 0th, 25th, 50th, 75th, and 100th answer length percentiles. The Pareto frontier is indicated by black dashed lines.

In Figure 15, we leverage the observation that incorrect answers are typically longer to design a simple early-stopping strategy, stopping generation once a specified answer length threshold is reached. For each model, we evaluate five thresholds corresponding to the 0th, 25th, 50th, 75th, and 100th answer length percentiles. We find that this straightforward strategy does not shift the Pareto frontier, as the reduction in inference cost comes at the expense of a notable drop in accuracy. Nevertheless, investigating more advanced approaches, such as behavior-conditioned inference (Didolkar et al.,

2025) or calibration-based abstention methods (Gisserot-Boukhlef et al., 2024a), to reduce unnecessary generation costs represents a promising direction for future research.

## F.4    Increasing Maximum Generation Length

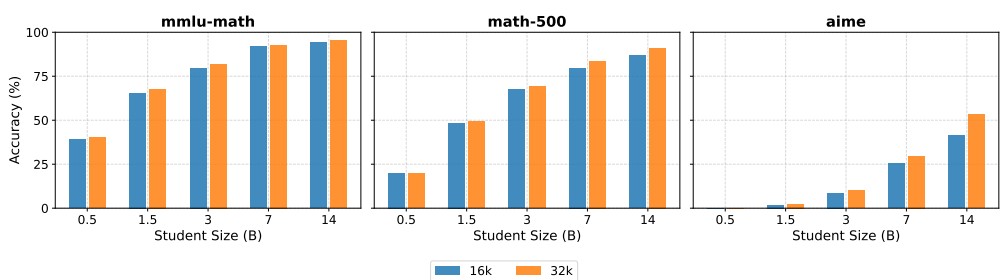

Figure 16: Impact of increasing maximum generation length (from 16,384 to 32,768 tokens) on downstream performance across `mmlu-math`, `math-500`, and `aime`.

Interestingly, Figure 16 shows that certain mathematical tasks benefit from increased generation length in the reasoning setting. In this experiment, models are allowed to generate up to 32,768 tokens, compared to the 16,384-token length used during training. This provides insight into why simple early-stopping strategies may fail, as some tasks require more tokens to produce correct answers. It also demonstrates that reasoning models can extrapolate well beyond the lengths on which they are trained, a behavior that could be further explored in future work.

## F.5    Inference Cost Scaling Trends

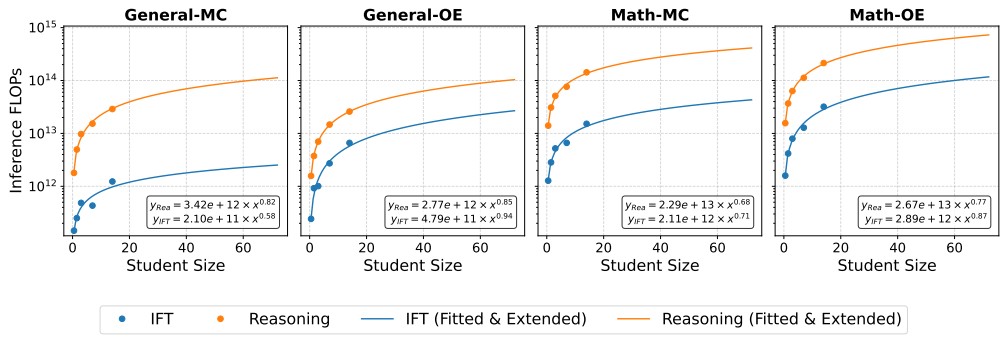

Figure 17: Inference FLOPs versus student model size for IFT and reasoning-style training. Points indicate the average inference FLOPs for each task category, while the curves show the corresponding log-linear scaling trends.

In Figure 17, we fit log-linear curves to inference FLOPs as a function of model size across task categories, assuming power-law relationships of the form $y = \alpha x^{\beta}$. The corresponding scaling coefficients are reported in each subplot. For General-OE, Math-MC, and Math-OE, the exponents $\beta$ are closely aligned ($\beta_{\text{IFT}} \approx \beta_{\text{Rea}} + 0.10$), slightly favoring $\beta_{\text{Rea}}$. This is consistent with Figure 8, where reasoning answers shorten slightly faster than IFT answers as model size increases. In contrast, for General-MC tasks, reasoning models display larger scaling coefficients than IFT models, indicating that the higher computational cost, combined with only marginal performance gains, limits the improvement observed on these tasks.

## G    TASK-LEVEL RESULTS

Figure 18, Figure 19, Figure 20, Figure 21,Figure 22, Figure 23, Figure 24 and Figure 25 present the task-level versions of the aggregated results shown in Figure 2, Figure 3, Figure 4, Figure 5, Figure 6, Figure 7 and Figure 8, respectively.

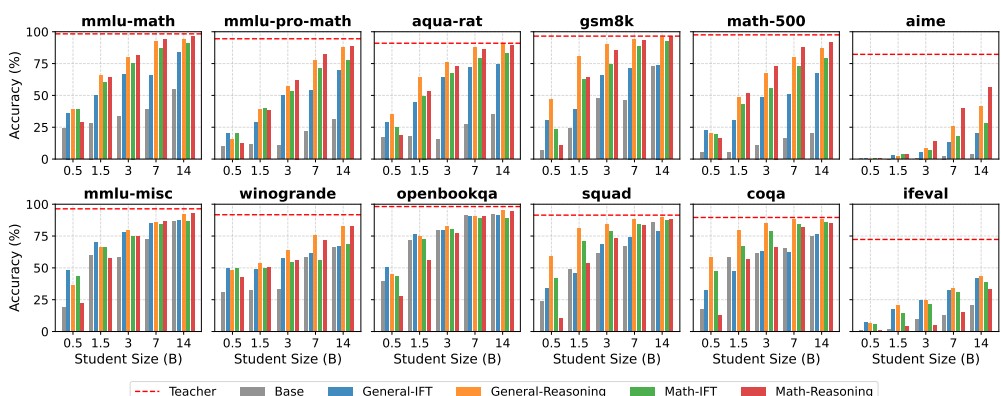

Figure 18: Task-level downstream performance of mono-phasic models.

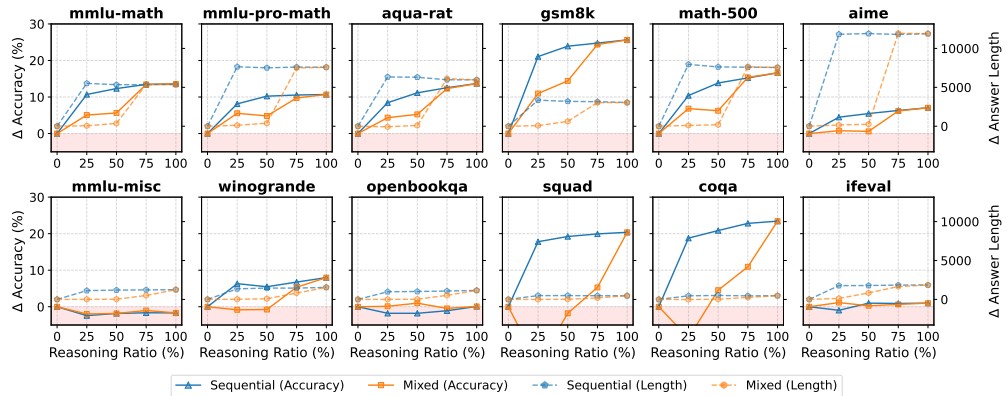

Figure 19: Task-level comparison of sequential and mixed training scenarios across varying reasoning ratios.

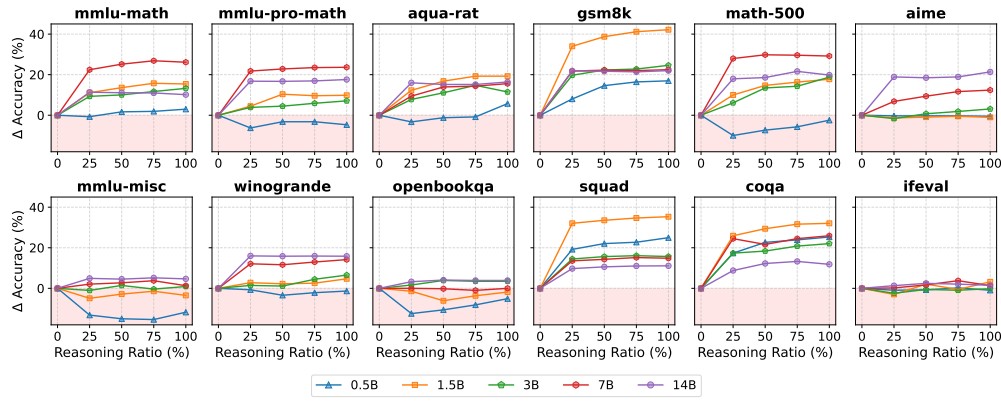

Figure 20: Task-level impact of the reasoning ratio on downstream performance.

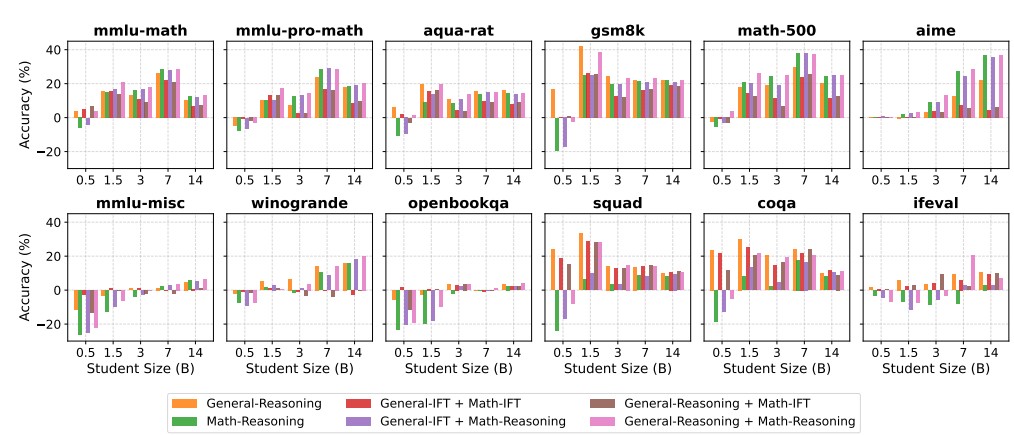

Figure 21: Task-level downstream performance of math-adapted models.

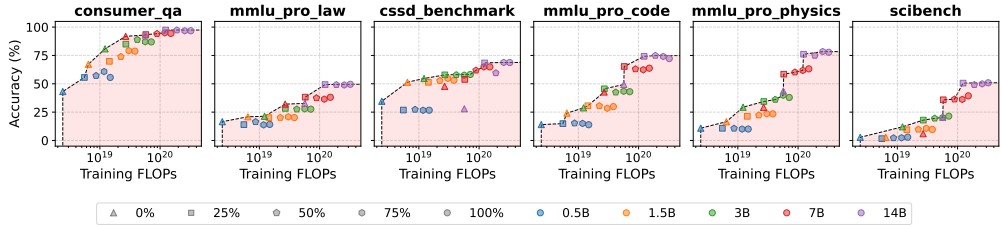

Figure 22: Task-level accuracy versus training FLOPs for models trained with IFT (0%), reasoning-style data (100%), and sequential reasoning ratios of 25%, 50%, and 75%.

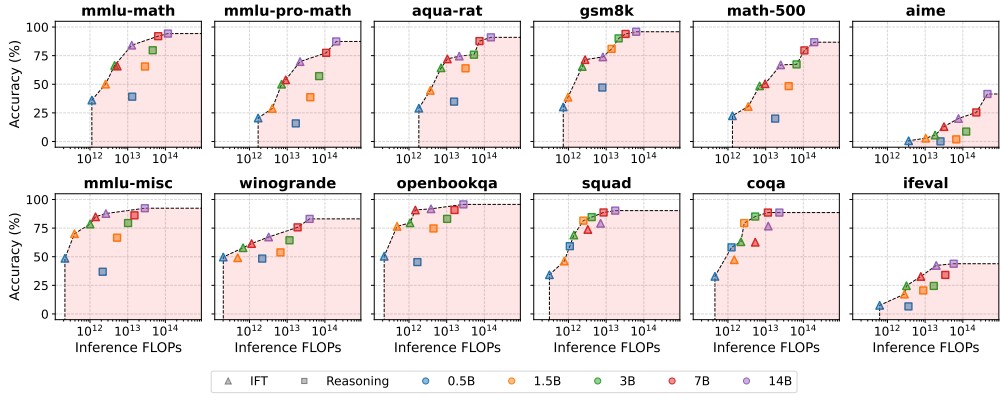

Figure 23: Task-level accuracy versus inference FLOPs for models trained with IFT and reasoning-style data.

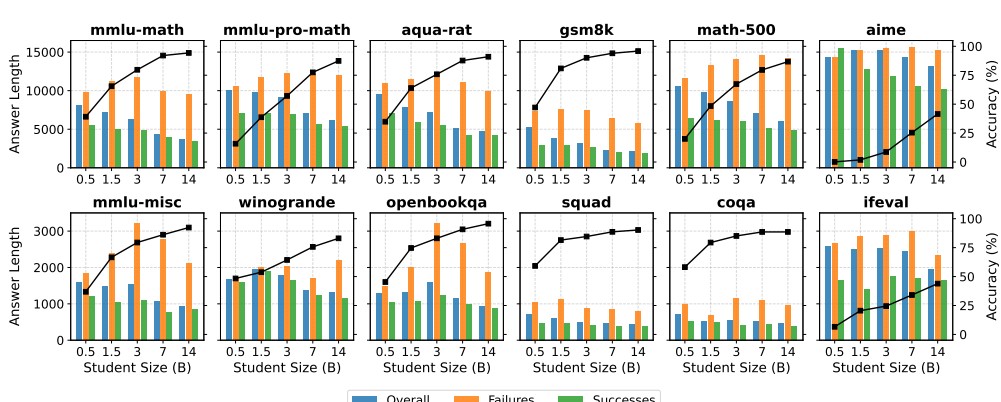

Figure 24: Task-level answer length analysis across student sizes and correctness in reasoning models.

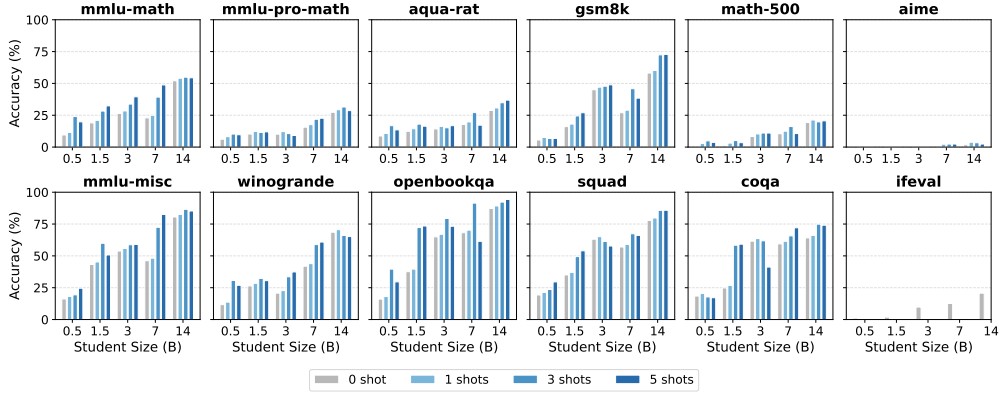

Figure 25: Performance evaluation of base student models in 0-shot inference (gray) against 1, 3, and 5-shot few-shot prompting (blue gradient). To mitigate the limited instruction-following capability of base student models, we report their performance in a three-shot setting in this paper.

