# OpenReview forum: "When Does Reasoning Matter? A Controlled Study of Reasoning’s Contribution to Model Performance"
_ICLR.cc/2026/Conference — Submitted to ICLR 2026_

### Official Review · Reviewer_6hA6 · 2025-10-31

**Soundness:** 3
**Presentation:** 3
**Contribution:** 2
**Rating:** 4
**Confidence:** 4

**Summary:**

This paper presents a thorough controlled study on when and how explicit reasoning via reasoning-augmented distillation, which improves large language model (LLM) performance compared to standard Instruction Fine-Tuning (IFT). Leveraging a synthetic dataset of paired IFT/reasoning teacher outputs across math-centric and general domains, the authors analyze accuracy, scaling, and resource efficiency over 12 benchmarks and a range of model sizes.

**Strengths:**

1. The paper evaluates over 12 benchmarks spanning multiple-choice and open-ended tasks, including specialized math and general-purpose domains, offering rich coverage. The study spans five model sizes (0.5B-14B) and multiple training protocols.
2. The synthetic data distillation setup carefully isolates the impact of reasoning traces from model/data/compute confounders, streamlining attribution. Training and evaluation use only teacher-generated paired data for apples-to-apples comparison, a significant methodological strength.

**Weaknesses:**

1. The evaluation relies exclusively on LLM-as-a-judge using Llama-3_1-Nemotron-Ultra-253B-v1. No human annotation or qualitative error analysis is provided.
2. While the study spans both math-specialized and general tasks, there is still a notable focus on math-centric reasoning. Other challenging reasoning domains (coding, legal, scientific, symbolic reasoning) are not substantially covered. This limits the generalizability of conclusions to all kinds of reasoning.
3. The manuscript briefly discusses sequential versus mixed training protocols and the impact of reasoning ratio, but deeper statistical analysis and explanations for the observed instabilities (such as the abrupt mode-switching in mixed training) are limited to observations rather than mechanistic investigation. For instance, why does accuracy sharply degrade beyond certain data ratios, and is this connected to the optimization landscape?

**Questions:**

See Weaknesses for details.

---

> ### Author Response · Authors · 2025-11-26
>
> We thank the reviewer for their positive assessment of our work. We are particularly grateful for their recognition of our rich empirical coverage and the methodological strength of our controlled distillation setup.
>
> Regarding the consensus **feedback of additional teacher-student pairs and additional reasoning intensive tasks, we kindly invite the reviewer to refer to the General Response above.**
>
> - **Explanation about mixed training observed instabilities:** We thank the reviewer for this question and for valuing the analysis in Figure 3. The abrupt shift in model behavior around a 50% reasoning ratio is driven by the inversion of the data distribution (switching from an IFT-dominant to a reasoning-dominant regime), which leads the model to default to a "reasoning mode" for all queries. While advanced techniques, such as data classification to gauge question difficulty, could help selectively activate reasoning capabilities, they often rely on external LLM-based systems [1, 2]. Introducing such systems would result in a less controlled experimental setup and is therefore beyond the scope of the current work. We plan to explore these data filtering techniques in future work (discussed in section Appendix A).
>
> We thank the reviewer again for their detailed feedback, which we think has significantly improved the quality of our manuscript. We hope we have addressed all their concerns and that they will consider raising their score. We would be happy to provide any further clarification and welcome the reviewer to ask any supplementary questions if needed.
>
> [1] Text-based question difficulty prediction: A systematic review of automatic approaches [2] Predicting difficulty and discrimination of natural language questions

---

### Official Review · Reviewer_hm41 · 2025-11-01

**Soundness:** 3
**Presentation:** 3
**Contribution:** 3
**Rating:** 6
**Confidence:** 3

**Summary:**

This paper presents a systematic empirical study on the role of chain-of-thought–like reasoning supervision versus instruction-following tuning (IFT) in LLM distillation. Using a controlled “paired IFT vs reasoning answer” setup from the same teacher, the authors investigate when and how reasoning traces improve downstream performance across (1) model scales (0.5B–14B), (2) training recipes (mono-phasic, mixed, sequential), (3) domains (general vs math), and (4) training/inference cost. Key findings include: 1) Reasoning improves performance primarily for larger models and open-ended/mathematical tasks; 2) IFT is generally Pareto-optimal in compute efficiency, whereas reasoning becomes competitive only at larger scales; 3) Mixed training exhibits occasional synergies but instability; sequential IFT to reasoning does not typically help; 4) Domain-specific math adaptation benefits ≥1.5B models but leads to forgetting for smaller ones.

**Strengths:**

1. Strong experimental rigor & scale: systematic evaluation across model sizes, training stages, domains, and compute budgets.
2. Controlled setup using paired reasoning vs non-reasoning teacher outputs significantly reduces confounding common in prior work.
3. Experimental findings: Reasoning supervision helps primarily for larger models and open-ended tasks. Instruction tuning remains a compute-efficient default baseline. Moderate reasoning proportions may be optimal in some cases.

**Weaknesses:**

1. Potential confounding factors remain. Although the work aims to isolate the causal effect of reasoning traces, several design choices may still introduce confounds. For example, the teacher uses different decoding hyperparameters (temperature / nucleus sampling) for IFT vs. reasoning traces, and student/base evaluation settings differ (zero-shot vs three-shot). These may bias learning dynamics in favor of one supervision format.
2. Focus is distillation-only. The conclusions emphasize “reasoning becomes valuable at scale,” but all models are distilled students trained from scratch. It is unclear whether conclusions generalize to: 1) RL-trained models; 2) long-trained base models (beyond 14B); 3) architectures other than Qwen.

**Questions:**

see weakness

---

> ### Author Response · Authors · 2025-11-26
>
> We thank the reviewer for their positive assessment of our experimental rigor and scale, particularly highlighting how our controlled, paired-output setup effectively isolates the impact of supervision format.
>
> Regarding the consensus **feedback of additional teacher-student pairs, we kindly invite the reviewer to refer to the General Response above.**
>
> - **Question about decoding hyperparameters:** To ensure optimal data quality and to avoid degrading the output quality of one format relative to the other we utilized the specific sampling parameters recommended by the developers of each teacher model ensuring that all training data is generated under optimal conditions.
>
> - **Few shots setting for base model:** For completeness, we have added the performance of base models across 0, 1, 3, and 5-shot settings in Appendix G, Figure 22. In the main figures, we reported base model performance in a few-shot setting [4] following standard practices in similar studies [1, 2, 3] for base models.
>
> - **Trained models with RL methods:** While frontier labs utilize RL for their largest models, technical reports from Qwen3 [2], DeepSeek-R1 [5] and Magistral [6] establish distillation as the standard for smaller models in the same family. These works demonstrate that smaller models trained solely via RL fail to match the performance and efficiency of those distilled from larger teachers. Consequently, our use of distillation accurately reflects standard practices for the model sizes evaluated in this study.
>
> - **Trained models beyond 14B:** Training models larger than 14B would have required moving directly to Qwen-32B or Gemma-27B, due to the gap in intermediate sizes. We attempted training Qwen-32B, but the 16K reasoning context required Sequence Parallelism, severely slowing training. A single run on 32 H100 GPUs took \~160 hours (>5,000 GPU-hours). Consequently, performing the 15 runs required for our full analysis would have totalled \~75,000 GPU-hours (\~$75k), which was computationally impractical for our setup.
>
> We thank the reviewer again for the detailed feedback, which we think has significantly improved the quality of our manuscript. We hope we have addressed all their concerns and that they will consider raising their score. We would be happy to provide any further clarification and welcome the reviewer to ask any supplementary questions if needed.
>
> [1] The Llama 3 Herd of Models [2] Qwen3 Technical Report [3] Gemma 3 Technical Report [4] Language Models are Few-Shot Learners [5] DeepSeek-R1: Incentivizing Reasoning Capability in LLMs via Reinforcement Learning [6] Magistral

---

### Official Review · Reviewer_tXHM · 2025-11-01

**Soundness:** 3
**Presentation:** 3
**Contribution:** 2
**Rating:** 4
**Confidence:** 4

**Summary:**

This paper investigates when explicit chain-of-thought (CoT) reasoning benefits LLM performance through a large-scale controlled distillation study. Using a single teacher (Qwen3-235B) generating paired IFT and reasoning responses, the authors train student models (0.5B-14B) on general and math-centric data across 12 benchmarks. Key findings: (1) reasoning consistently improves performance, especially on open-ended and math tasks, often matching larger IFT models; (2) IFT remains Pareto-optimal for training/inference costs; (3) reasoning benefits increase with model scale, breaking IFT performance plateaus at 7B+ parameters.

**Strengths:**

**Comprehensive Empirical Coverage**: The controlled distillation framework is the paper's main strength. By having a single teacher generate paired IFT/reasoning outputs for identical prompts, the authors cleanly isolate the effect of supervision format while controlling for data domain, teacher capacity, and other confounders. This represents a significant methodological advance over prior work comparing models trained on different data mixtures. The study spans 5 model sizes (0.5B-14B), 2 data distributions (general/math), 12 benchmarks across 4 task categories (General-MC/OE, Math-MC/OE), multiple reasoning ratios (0%, 25%, 50%, 75%, 100%), and two training regimes (sequential/mixed). The 70k GPU-hour investment and 1.6M training pairs demonstrate serious experimental commitment. The task-level granularity (Appendix G) adds valuable nuance.

**Practical Cost-Benefit Analysis**: Sections 4.1-4.2 provide actionable guidance by analyzing training/inference FLOPs versus accuracy. The Pareto frontier analysis (Figures 6-7) clearly shows IFT's efficiency advantage while demonstrating reasoning's value at larger scales. The finding that intermediate reasoning ratios (25-75%) achieve Pareto optimality is practically valuable. The analysis of answer length correlating with errors (Figure 8) offers insight into inference cost drivers.

**Weaknesses:**

1. Lack of formalized scaling laws and reusable framework. Despite extensive ablations across ~50+ configurations, the paper fails to synthesize findings into predictive scaling laws or actionable decision frameworks. For example, an unified equations: The paper observes that reasoning benefits scale with model size and vary by task, but doesn't formalize these relationships (e.g., `Accuracy_gain(N, Task) = f(parameters)`). Researchers cannot predict whether reasoning will help for their specific setup without replicating experiments.
2. Single teacher architecture and No cross-teacher validation: All data comes from Qwen3-235B. Teacher-specific biases, reasoning styles, or artifacts may not generalize to other models (GPT, Claude, Llama families). Testing students trained on Teacher A's reasoning traces with Teacher B's evaluation would strengthen claims.
3. No RL baseline and limited task domains: Frontier labs use RL (GRPO, PPO) for reasoning models. Comparing pure distillation to RL-based reasoning would contextualize practical relevance. Focus on math/general tasks ignores code (brief mention but no systematic evaluation), science, legal reasoning, creative writing, etc.

**Questions:**

1. How sensitive are results to teacher choice? Would a Llama-405B or GPT-4 teacher yield similar task sensitivities and scaling trends?
2. How does your best distilled reasoning model (14B, 100% ratio) compare to a 14B model trained with RL (GRPO/PPO) on the same tasks?
3. Can you provide fitted equations for the key relationships observed?

---

> ### Author Response · Authors · 2025-11-26
>
> We thank the reviewer for their comprehensive evaluation and feedback. We are particularly grateful for their recognition of our controlled distillation framework as a significant methodological advance, and their appreciation of our cost-benefit analysis (Pareto frontiers) as offering actionable guidance for practitioners.
>
> Regarding the consensus **feedback of additional teacher-student pairs, we kindly invite the reviewer to refer to the General Response above.**
>
> - **Teacher Selection:** We excluded GPT-4 due to Terms of Service restrictions on distillation, and private API models due to reproducibility concerns and prohibitive costs (~$368k for Claude 4.5). To address the need for additional teachers, we conducted additional experiments with Nemotron-Super-49B-v1.5, a model achieving comparable performance of Llama 400B
>
> - **How practitioners can predict whether reasoning will help for their specific setup:** Unlike pre-training, no unified scaling equations exist for downstream tasks [1, 2]. We therefore avoided extrapolating a potentially misleading scaling law from a single task and instead conducted a large-scale empirical analysis (~50+ configurations). We also released all models from our controlled setup, allowing practitioners to assess the impact of reasoning on their tasks at minimal cost (inference only). We show this extensibility in Appendix F.1 by evaluating 6 new reasoning-intensive tasks.
>
> - **Frontier labs use RL (GRPO, PPO) for reasoning models:** While frontier labs utilize RL for their largest models, technical reports from DeepSeek-R1 [3], Qwen3 [4], and Magistral [5] establish distillation as the standard for smaller models in the same family. These works demonstrate that smaller models trained solely via RL fail to match the performance and efficiency of those distilled from larger teachers. Consequently, our use of distillation accurately reflects standard practices for the model sizes evaluated in this study and above.
>
> We thank the reviewer again for their detailed feedback, which we think has significantly improved the quality of our manuscript. We hope we have addressed all their concerns and that they will consider raising their score. We would be happy to provide any further clarification and welcome the reviewer to ask any supplementary questions if needed.
>
> [1] Scaling Laws Are Unreliable for Downstream Tasks: A Reality Check [2] How predictable is language model benchmark performance? [3] DeepSeek-R1: Incentivizing Reasoning Capability in LLMs via Reinforcement Learning [4] Qwen3 Technical Report [5] Magistral

---

### Official Review · Reviewer_yQMt · 2025-11-01

**Soundness:** 2
**Presentation:** 2
**Contribution:** 2
**Rating:** 4
**Confidence:** 3

**Summary:**

The paper presents a controlled, large-scale study comparing instruction fine-tuning (IFT) versus explicit “reasoning”-style supervision across model scales (0.5B–14B), task families (general vs. math), and answer formats (multiple-choice vs. open-ended). Using a single teacher to generate paired IFT and reasoning outputs for the same prompts (1.6M pairs), the authors disentangle supervision format effects from confounders and analyze accuracy–efficiency tradeoffs (training and inference FLOPs). Key findings: (i) reasoning consistently helps, especially on math and open-ended tasks, sometimes letting smaller students match much larger IFT models; (ii) IFT remains Pareto-optimal for cost, while reasoning becomes increasingly valuable at larger scales; (iii) mixed IFT+reasoning has limited stability benefits; (iv) longer generations correlate with higher error rates.

**Strengths:**

Clean experimental design / strong controls. A single-teacher, paired-output distillation pipeline isolates the causal effect of supervision format, avoiding data-mixture and RL confounds. The scale (1.6M pairs; 12 benchmarks) is substantial and well-documented.

Actionable takeaways for practitioners. Clear guidance on when to prefer reasoning vs. scaling IFT, with explicit Pareto analyses over training and inference FLOPs.

Nuanced findings by task family and format. Reasoning helps most on math and open-ended; gains on general MC are modest/inconsistent. Results also cover domain adaptation (math).

**Weaknesses:**

Judge-model dependence. Heavy reliance on LLM-as-a-judge (Nemotron-Ultra-253B) introduces potential bias and circularity. Robustness to alternative judges or human spot-checks is not reported.

Single-teacher bias. Using one teacher (Qwen3-235B-A22B) risks transferring teacher-specific inductive biases. Ablations with a second teacher (or ensemble) would strengthen causal claims.

Mixed-training instability. Useful observation, but practical guidance is thin—why instability occurs and how to stabilize data mixing remains open.

**Questions:**

Judge robustness. How do results change with (a) a second judge model of a different family/size, set to validate judge decisions?


Teacher diversity. Did the team try a secondary teacher (e.g., different architecture/training mix)? Even a smaller sample would clarify teacher-bias sensitivity.

---

> ### Author Response · Authors · 2025-11-26
>
> We thank the reviewer for the thorough review and valuable feedback. We are grateful that the reviewer highlighted our rigorous experimental design, the practical utility of our findings, and our detailed analysis across diverse benchmarks.
>
> Regarding the consensus **feedback on Judge-model dependence and Single-teacher bias, we kindly invite the reviewer to refer to the General Response above.**
>
> - **Details concerning the instability discovered with mix shuffling strategies:** The abrupt shift in model behavior around a 50% reasoning ratio is driven by the inversion of the data distribution (switching from an IFT-dominant to a reasoning-dominant regime), which leads the model to default to a "reasoning mode" for all queries. While advanced techniques, such as data classification to gauge question difficulty, could help selectively activate reasoning capabilities, they often rely on external LLM-based systems [1, 2] Introducing such systems would result in a less controlled experimental setup and is therefore beyond the scope of the current work. We plan to explore these data filtering techniques in future work (discussed in section Appendix A).
>
> We thank the reviewer again for the detailed feedback, which we think has significantly improved the quality of our manuscript. We hope we have addressed all their concerns and that they will consider raising their score. We would be happy to provide any further clarification and welcome the reviewer to ask any supplementary questions if needed.
>
> [1] Text-based question difficulty prediction: A systematic review of automatic approaches [2] Predicting difficulty and discrimination of natural language questions

---

### Author Response · Authors · 2025-11-26
**General Response: Summary of New Experiments and Manuscript Updates**

We thank the reviewers for their thorough review and positive feedback. We are grateful that reviewers found our work with **“strong experimental rigor & scale” (hm41)**, noting that it **"represents a significant methodological advance over prior work” (tXHM)** by **“carefully isolates the impact of reasoning traces from model/data/compute confounders” (6hA6)** and leads to **"actionable takeaways for practitioners” (yQMt)**, strengthened by the reproducibility enabled by our open-sourced models and code.

Beyond the strengths, two limitations were identified—judge diversity and the experimental scope of our findings. To address these issues, we conducted a large collection of additional experiments. **These new experiments validate and further strengthen the original results.** The new findings have been incorporated into the revised manuscript, and we discuss them collectively in the consolidated response below.

# New Judge Models and Majority Voting (reviewers yQMt, tXHM, 6hA6)
To strengthen robustness, we incorporated two additional judges (gpt-oss-120b and Llama-3.3-70B-Instruct) and employed majority voting [1, 2] across all model answers. This setup increases reliability and confirms the trends reported in our paper. We also emphasize that judges evaluate semantic equivalence against explicit golden labels rather than performing open-ended assessment. This approach prevents false negatives due to formatting discrepancies (e.g., '32$' vs '32.00'), a common failure mode of the exact-match method, ensuring fairness and manual verification. We have updated the manuscript plots confirming our findings, refined 'LLM-as-a-Judge' section, and details on majority voting in Appendix D.3.

# Additional Experiments (reviewers yQMt, tXHM, hm41, 6hA6):
- **Exploring Alternative Teacher-Student Configurations (reviewers yQMt, tXHM, hm41):** To address the concern regarding teacher diversity, we conducted an additional experiment using a new pair of teacher (Nemotron-Super-49B-v1.5) and student family (Gemma 3: 1B, 4B, and 12B) trained on 200K samples of newly generated data. The evaluation of this new teacher-student configuration exhibits patterns similar to those observed in the core manuscript. This reinforces our previous observations and provides compelling evidence that our conclusions are robust to teacher and student choice. We have updated the manuscript with a new section, "Exploring Alternative Teacher-Student Configurations" (Appendix F.2).
- **Generalization Across Reasoning Domains (reviewers tXHM, 6hA6):** We extended our analysis on six challenging reasoning benchmarks spanning three fields: Law, Code, and Physics. For each domain, we evaluated one multiple-choice task and one open-ended generation task. These diverse tasks exhibit trends similar to those reported in the main paper (aligning particularly well with the math tasks), confirming that our findings are consistent across different reasoning domains. These new results are detailed in Appendix F.1 (Figures 9, 10, and 11).

We thank reviewers again for their detailed feedback, which we think has significantly improved the quality of our manuscript. **We have incorporated all suggested ablations and analyses into the updated paper** and remain available for any further clarification.

[1] Predicting judicial decisions: A statistically rigorous approach and a new ensemble classifier
[2] JurEE not Judges: safeguarding llm interactions with small, specialised Encoder Ensembles

---

### Meta-Review · Area_Chair_T3BW · 2026-01-03

**Summary:**

This paper presents a large-scale controlled study comparing Instruction Fine-Tuning (IFT) with reasoning-based distillation, utilizing a synthetic dataset to analyze performance and cost trade-offs across model scales (0.5B–14B).
However, the decision to reject stems from the consensus that the study’s scope is too narrow to properly characterize the emergence of reasoning capabilities in modern LLMs. Specifically, the exclusive focus on distillation—ignoring Reinforcement Learning (RL) baselines—limits the applicability of the findings to the current state-of-the-art, and the lack of formal scaling laws or mechanistic insights into training instabilities restricts the work's impact despite its experimental rigor.

**Reviewer Concerns:**

The authors mitigated concerns regarding evaluation robustness by incorporating diverse judge models (GPT-OSS, Llama 3.3) and extending the domain coverage to Law and Code.

However, the critical concern regarding the lack of comparison against RL-trained models remains outstanding, as reviewers argue distillation alone does not capture the full dynamics of reasoning emergence found in frontier models.
Additionally, the mechanistic explanation for the observed instability in mixed training regimes remains qualitative and lacks the theoretical depth or formal scaling laws expected for a controlled scientific study.

**Reviewer Scores:**

Reviewers yQMt, tXHM, and 6hA6 are likely to retain their scores; while they would appreciate the additional experiments and new teacher configurations, the fundamental limitation of excluding RL baselines prevents the work from meeting the bar for a clear acceptance.

---

### Decision · Program_Chairs · 2026-01-26

Reject